# OpenCoS: Contrastive Semi-supervised Learning for Handling Open-set Unlabeled Data

## Abstract

Modern semi-supervised learning methods conventionally assume both labeled and unlabeled data have the same class distribution. However, unlabeled data may include *out-of-class* samples in practice; those that cannot have one-hot encoded labels from a closed-set of classes in label data, *i.e.*, unlabeled data is an *open-set*. In this paper, we introduce *OpenCoS*, a method for handling this realistic semi-supervised learning scenario based on a recent framework of contrastive learning. One of our key findings is that out-of-class samples in the unlabeled dataset can be identified effectively via (unsupervised) contrastive learning. OpenCoS utilizes this information to overcome the failure modes in the existing state-of-the-art semi-supervised methods, *e.g.*, ReMixMatch or FixMatch. In particular, we propose to assign soft-labels for out-of-class samples using the representation learned from contrastive learning. Our extensive experimental results show the effectiveness of OpenCoS, fixing the state-of-the-art semi-supervised methods to be suitable for diverse scenarios involving open-set unlabeled data. The code will be released.

## 1 Introduction

Despite the recent success of deep neural networks with large-scale labeled data, many real-world scenarios suffer from expensive data acquisition and labeling costs. This has motivated the community to develop *semi-supervised learning* (SSL; Grandvalet & Bengio 2004; Chapelle et al. 2009), *i.e.*, by further incorporating unlabeled data for training. Indeed, recent SSL works (Berthelot et al., 2019; 2020; Sohn et al., 2020) demonstrate promising results on several benchmark datasets, as they could even approach the performance of fully supervised learning using only a small number of labels, *e.g.*, 93.73% accuracy on CIFAR-10 with 250 labeled data (Berthelot et al., 2020).

However, SSL methods often fail to generalize when there is a mismatch between the class-distributions of labeled and unlabeled data (Oliver et al., 2018; Chen et al., 2020c; Guo et al., 2020), *i.e.*, when the unlabeled data contains *out-of-class* samples, whose ground-truth labels are not contained in the labeled dataset (as illustrated in Figure 1(a)). In this scenario, various label-guessing techniques used in the existing SSL methods may label those out-of-class samples incorrectly, which in turn significantly harms the overall training through their inner-process of entropy minimization (Grandvalet & Bengio, 2004; Lee, 2013) or consistency regularization (Xie et al., 2019; Sohn et al., 2020). This problem may largely hinder the existing SSL methods from being used in practice, considering the *open-set* nature of unlabeled data collected in the wild (Bendale & Boult, 2016).

**Contribution.** In this paper, we focus on a realistic SSL scenario, where unlabeled data may contain some unknown *out-of-class samples*, *i.e.*, there is a class distribution mismatch between labeled and unlabeled data (Oliver et al., 2018). Compared to prior approaches that have bypassed this problem by simply filtering out them with some heuristic detection scores (Nair et al., 2019; Chen et al., 2020c), the unique characteristic in our approach is to further leverage the information in out-of-class samples by assigning *soft-labels* to them: they may still contain some useful features for the in-classes.

Somewhat surprisingly, we found that a recent technique of *contrastive unsupervised learning* (Wu et al., 2018; He et al., 2020; Chen et al., 2020a) can play a key role for our goal. More specifically, we show that a pre-trained representation via contrastive learning, namely SimCLR (Chen et al., 2020a), on both labeled and unlabeled data enables us to design (a) an effective score for detecting out-of-class samples in unlabeled data, and (b) a systematic way to assign soft-labels to the detected out-of-class samples, by modeling *class-conditional likelihoods* from labeled data. Finally, we found (c) auxiliary

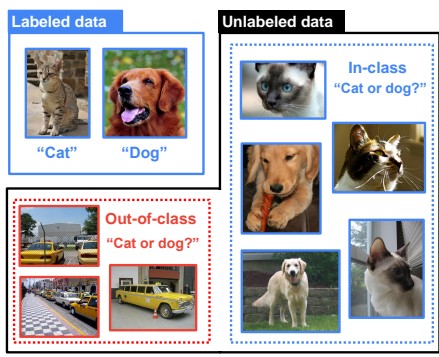
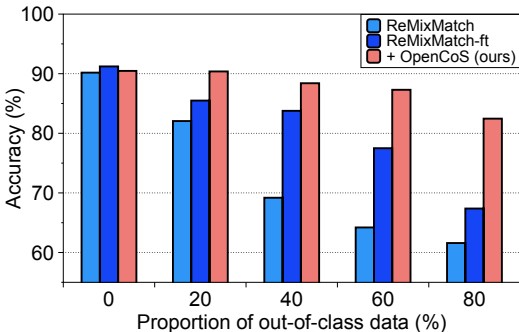

(a) Class-distribution mismatch

(b) Comparison of median test accuracy

Figure 1: (a) Illustration of an open-set unlabeled data under class-distribution mismatch in semi-supervised learning, *i.e.*, unlabeled data may contain unknown out-of-class samples. (b) Comparison of median test accuracy under varying proportions of out-of-class samples on the CIFAR-10 + TinyImageNet benchmark with 25 labels per class.

batch normalization layers (Xie et al., 2020) could further help to mitigate the class-distribution mismatch via decoupling batch normalization layers. We propose a generic SSL framework, coined *OpenCoS*, based on the aforementioned techniques for handling open-set unlabeled data, which can be integrated with any existing SSL methods.

We verify the effectiveness of the proposed method on a wide range of SSL benchmarks based on CIFAR-10, CIFAR-100 (Krizhevsky et al., 2009), and ImageNet (Deng et al., 2009) datasets, assuming the presence of various out-of-class data, *e.g.*, SVHN (Netzer et al., 2011) and TinyImageNet datasets. Our experimental results demonstrate that OpenCoS greatly improves existing state-of-the-art SSL methods (Berthelot et al., 2019; 2020; Sohn et al., 2020), not only by discarding out-of-class samples, but also by further leveraging them into training. We also compare our method to other recent works (Nair et al., 2019; Chen et al., 2020c; Guo et al., 2020) addressing the same class distribution mismatch problem in SSL, and again confirms the effectiveness of our framework, *e.g.*, we achieve an accuracy of 68.37% with 40 labels (just 4 labels per class) on CIFAR-10 with TinyImageNet as out-of-class, compared to DS³L (Guo et al., 2020) of 56.32%.

Overall, our work highlights the benefit of unsupervised representations in (semi-) supervised learning: such a *label-free* representation turns out to enhance model generalization due to its robustness on the novel, out-of-class samples.

## 2 PRELIMINARIES

### 2.1 SEMI-SUPERVISED LEARNING

The goal of *semi-supervised learning* for classification is to train a classifier $f : \mathcal{X} \to \mathcal{Y}$ from a *labeled dataset* $\mathcal{D}_l = \{x_l^{(i)}, y_l^{(i)}\}_{i=1}^{N_l}$ where each label $y_l$ is from a set of classes $\mathcal{Y} := \{1, \cdots, C\}$, and an *unlabeled dataset* $\mathcal{D}_u = \{x_u^{(i)}\}_{i=1}^{N_u}$ where each $y_u$ exists but is assumed to be unknown. In an attempt to leverage the extra information in $\mathcal{D}_u$, a number of techniques have been proposed, *e.g.*, entropy minimization (Grandvalet & Bengio, 2004; Lee, 2013) and consistency regularization (Sajjadi et al., 2016). In general, recent approaches in semi-supervised learning can be distinguished by the prior they adopt for the representation of unlabeled data: for example, the consistency regularization technique (Sajjadi et al., 2016) attempt to minimize the cross-entropy loss between any two predictions of different augmentations $t_1(x_u)$ and $t_2(x_u)$ from a given unlabeled sample $x_u$, jointly with the standard training for a labeled sample $(x_l, y_l)$:

$$\mathcal{L}_{\mathrm{SSL}}(x_l, x_u) := \mathbb{H}(y_l, f(x_l)) + \beta \cdot \mathbb{H}(f(t_1(x_u)), f(t_2(x_u))), \qquad (1)$$

where $\mathbb{H}$ is a standard cross-entropy loss for labeled data, and $\beta$ is a hyperparameter. Recently, several "holistic" approaches of various techniques (Zhang et al., 2018; Cubuk et al., 2019) have shown remarkable performance in practice, *e.g.*, MixMatch (Berthelot et al., 2019), ReMixMatch (Berthelot et al., 2020), and FixMatch (Sohn et al., 2020), which we mainly consider in this paper. We note that our scheme can be integrated with any recent semi-supervised learning methods.

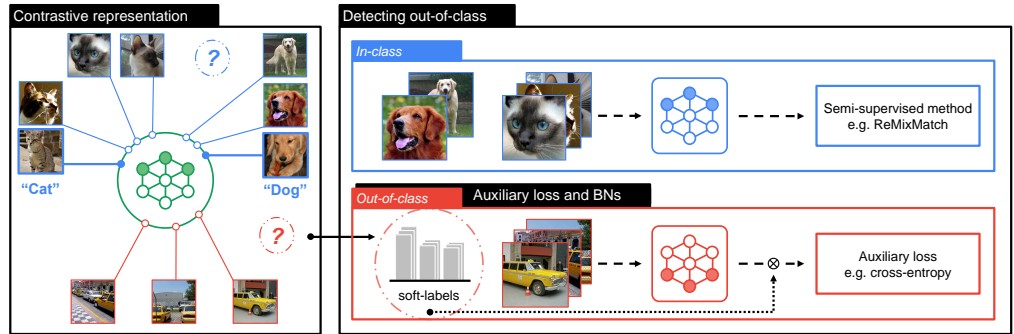

Figure 2: Overview of our proposed framework, *OpenCoS*. First, our method detects out-of-class samples based on contrastive representation. The out-of-class samples detected by OpenCoS are further utilized via an auxiliary loss with soft-labels generated from the representation, while the remaining in-class samples are used for standard semi-supervised methods. Also, the out-of-class samples pass through additional batch normalization layers to handle a class-distribution mismatch.

## 2.2 CONTRASTIVE REPRESENTATION LEARNING

Contrastive learning (Oord et al., 2018; Hénaff et al., 2019; He et al., 2020; Chen et al., 2020a) defines an unsupervised task for an encoder $f_e : \mathcal{X} \to \mathbb{R}^{d_e}$ from a set of samples $\{x_i\}$: assume that a "query" sample $x_q$ is given and there is a positive "key" $x_+ \in \{x_i\}$ that $x_q$ matches. Then the *contrastive loss* is defined to let $f$ to extract the necessary information to identify $x_+$ from $x_q$ as follows:

$$\mathcal{L}^{\text{con}}(f_e, x_q, x_+; \{x_i\}) := -\log \frac{\exp(h(f_e(x_q), f_e(x_+))/\tau)}{\sum_i \exp(h(f_e(x_q), f_e(x_i))/\tau)}, \tag{2}$$

where $h(\cdot, \cdot)$ is a pre-defined similarity score, and $\tau$ is a temperature hyperparameter. In this paper, we primarily focus on *SimCLR* (Chen et al., 2020a), a particular form of contrastive learning: for a given $\{x_i\}_{i=1}^N$, SimCLR first samples two separate data augmentation operations from a pre-defined family $\mathcal{T}$, namely $t_1, t_2 \sim \mathcal{T}$, and matches $(\tilde{x}_i, \tilde{x}_{i+N}) := (t_1(x_i), t_2(x_i))$ as a query-key pair interchangeably. The actual loss is then defined as follows:

$$\mathcal{L}^{\text{SimCLR}}(f_e; \{x_i\}_{i=1}^N) := \frac{1}{2N} \sum_{q=1}^{2N} \mathcal{L}^{\text{con}}(f_e, \tilde{x}_q, \tilde{x}_{(q+N) \bmod 2N}; \{\tilde{x}_i\}_{i=1}^{2N} \setminus \{\tilde{x}_q\}), \tag{3}$$

$$h^{\text{SimCLR}}(v_1, v_2) := \text{CosineSimilarity}(g(v_1), g(v_2)) = \frac{g(v_1) \cdot g(v_2)}{||g(v_1)||_2 ||g(v_2)||_2}, \tag{4}$$

where $g : \mathbb{R}^{d_e} \to \mathbb{R}^{d_p}$ is a 2-layer neural network called *projection header*. In other words, the SimCLR loss defines a task to identify a "semantically equivalent" sample to $x_q$ up to the set of data augmentations $\mathcal{T}$.

## 3 OPENCOS: A FRAMEWORK FOR OPEN-SET SEMI-SUPERVISED LEARNING

We consider semi-supervised classification problems involving $C$ classes. In addition to the standard assumption of semi-supervised learning (SSL), we assume that the unlabeled dataset $\mathcal{D}_u$ is *open-set*, *i.e.*, the hidden labels $y_u$ of $x_u$ may not be in $\mathcal{Y} := \{1, \cdots, C\}$. In this scenario, existing semi-supervised learning techniques may degrade the classification performance, possibly due to incorrect label-guessing procedure for those out-of-class samples. In this respect, we introduce *OpenCoS*, a generic method for detecting and labeling out-of-class unlabeled samples in semi-supervised learning. Overall, our key intuition is to utilize the unsupervised representation from *contrastive learning* (Wu et al., 2018; He et al., 2020; Chen et al., 2020a) to leverage such out-of-class samples in an appropriate manner. We present a brief overview of our method in Section 3.1, and describe how our approach, OpenCoS, can handle out-of-class samples in Section 3.2 and 3.3.

### 3.1 OVERVIEW OF OPENCOS

Recall that our goal is to train a classifier $f : \mathcal{X} \to \mathcal{Y}$ from a labeled dataset $\mathcal{D}_l$ and an *open-set* unlabeled dataset $\mathcal{D}_u$. Overall, OpenCoS aims to overcome the presence of out-of-class samples in $\mathcal{D}_u$ through the following procedure:

1. **Pre-training via contrastive learning.** OpenCoS first learns an unsupervised representation of $f$ via SimCLR[1] (Chen et al., 2020a), using both $\mathcal{D}_l$ and $\mathcal{D}_u$ without labels. More specifically, we learn the penultimate features of $f$, denoted by $f_e$, by minimizing the contrastive loss defined in (3). We also introduce a projection header $g$ (4), which is a 2-layer MLP as per (Chen et al., 2020a).

2. **Detecting out-of-class samples.** From a learned representation of $f_e$ and $g$, OpenCoS identifies an *out-of-class* unlabeled data $\mathcal{D}_u^{\text{out}}$ from the given data $\mathcal{D}_u = \mathcal{D}_u^{\text{in}} \cup \mathcal{D}_u^{\text{out}}$. This detection process is based on the similarity score between $\mathcal{D}_l$ and $\mathcal{D}_u$ in the representation space of $f_e$ and $g$ (see Section 3.2).

3. **Semi-supervised learning with auxiliary loss and batch normalization.** Now, one can use any semi-supervised learning scheme to train $f$ using $\mathcal{D}_l$ and $\mathcal{D}_u^{\text{in}}$, *e.g.*, ReMixMatch (Berthelot et al., 2020). In addition, OpenCoS minimizes an *auxiliary loss* that assigns a soft-label to each sample in $\mathcal{D}_u^{\text{out}}$, which is also based on the representation of $f_e$ and $g$ (see Section 3.3). Furthermore, we found maintaining auxiliary batch normalization layers (Xie et al., 2020) for $\mathcal{D}_u^{\text{out}}$ is beneficial to our loss as they mitigate the distribution mismatch arisen from $\mathcal{D}_u^{\text{out}}$.

Putting it all together, OpenCoS provides an effective and systematic way to detect and utilize out-of-class data for semi-supervised learning. Due to its simplicity, our framework can incorporate the most recently proposed semi-supervised learning methods (Berthelot et al., 2019; 2020; Sohn et al., 2020) and improve their performance in the presence of out-of-class samples. Figure 2 illustrates the overall training scheme of OpenCoS.

## 3.2 DETECTION CRITERION OF OPENCOS

For a given labeled dataset $\mathcal{D}_l$ and an open-set unlabeled dataset $\mathcal{D}_u$, we aim to detect a subset of the unlabeled training data $\mathcal{D}_u^{\text{out}} \subseteq \mathcal{D}_u$ whose elements are out-of-class, *i.e.*, $y_u \notin \mathcal{Y}$. A standard way to handle this task is to train a confident-calibrated classifier using $\mathcal{D}_l$ (Hendrycks & Gimpel, 2017; Liang et al., 2018; Lee et al., 2018a;b; Hendrycks et al., 2019a;b; Bergman & Hoshen, 2020; Tack et al., 2020). However, such methods typically assume a sufficient number of in-class samples (*i.e.*, large $\mathcal{D}_l$), which does not hold in our case to the label-scarce nature of SSL. This motivates us to consider a more suitable approach which leverages the open-set unlabeled dataset $\mathcal{D}_u$ for contrastive learning. Then, OpenCoS utilizes the labeled dataset $\mathcal{D}_l$ to estimate the class-wise distributions of (pre-trained) embeddings, and use them to define a detection score for $\mathcal{D}_u$.

We assume that an encoder $f_e : \mathcal{X} \to \mathbb{R}^{d_e}$ and a projection header $g : \mathbb{R}^{d_e} \to \mathbb{R}^{d_p}$ pre-trained via SimCLR on $\mathcal{D}_l \cup \mathcal{D}_u$. Motivated by the similarity metric used in the pre-training objective of SimCLR (4), we propose a simple yet effective detection score $s(x_u)$ for unlabeled input $x_u$ based on the cosine similarity between $x_u$ and *class-wise prototypical representations* $\{v_c\}_{c=1}^{C}$ obtained from $\mathcal{D}_l$. Namely, we first define a *class-wise similarity score*[2] $\text{sim}_c(x_u)$ for each class $c$ as follows:

$$v_c\left(\mathcal{D}_l; f_e, g\right) := \frac{1}{N_l^c} \sum_i \mathbf{1}_{y_l^{(i)}=c} \cdot g(f_e(x_l^{(i)})), \quad \text{and} \tag{5}$$

$$\text{sim}_c\left(x_u; \mathcal{D}_l, f_e, g\right) := \text{CosineSimilarity}(g(f_e(x_u)), v_c), \tag{6}$$

where $N_l^c := |\{(x_l^{(i)}, y_l^{(i)})|y_l^{(i)} = c\}|$ is the sample size of class $c$ in $\mathcal{D}_l$. Then, our detection score $s(x_u)$ is defined by the maximal similarity score between $x_u$ and the prototypes $\{v_c\}_{c=1}^{C}$:

$$s(x_u) := \max_{c=1,\cdots,C} \text{sim}_c\left(x_u\right). \tag{7}$$

In practice, we use a pre-defined threshold $t$ for detecting out-of-class samples in $\mathcal{D}_u$, *i.e.*, we detect a given sample $x_u$ as out-of-class if $s(x_u) < t$. In our experiments, we found an empirical value of $t := \mu_l - 2\sigma_l$ performs well across all the datasets tested, where $\mu_l$ and $\sigma_l$ are mean and standard deviation computed over $\{s(x_l^{(i)})\}_{i=1}^{N_l}$, respectively, although more tuning of $t$ could further improve the performance. Further analysis of our detection threshold can be found in Appendix B.4.

---

[1]Nevertheless, our framework is not restricted to a single method of SimCLR; it is easily generalizable to other contrastive learning methods (Hénaff et al., 2019; He et al., 2020; Chen et al., 2020b).

[2]In this work, we adopt the well-known cosine similarity to define our score, but there can be other designs as long as it represents class-wise similarity (Chen et al., 2020d; Vinyals et al., 2016; Snell et al., 2017).

### 3.3 Auxiliary loss and batch normalization of OpenCoS

Based on the detection criterion defined in Section 3.2, the open-set unlabeled dataset $\mathcal{D}_u$ can be split into (a) the *in-class* unlabeled dataset $\mathcal{D}_u^{\text{in}}$ and (b) the *out-of-class* unlabeled dataset $\mathcal{D}_u^{\text{out}}$. The labeled dataset $\mathcal{D}_l$ and $\mathcal{D}_u^{\text{in}}$ are now used to train the classifier $f$ using any existing semi-supervised learning method (Berthelot et al., 2019; 2020; Sohn et al., 2020).

In addition, we propose to further utilize $\mathcal{D}_u^{\text{out}}$ via an auxiliary loss that assigns a soft-label to each $x_u^{\text{out}} \in \mathcal{D}_u^{\text{out}}$. More specifically, for any semi-supervised learning objective $\mathcal{L}_{\text{SSL}}(x_l, x_u^{\text{in}}; f)$, we consider the following loss:

$$\mathcal{L}_{\text{OpenCoS}} = \mathcal{L}_{\text{SSL}}(x_l, x_u^{\text{in}}; f) + \lambda \cdot \mathbb{H}(q(x_u^{\text{out}}), f(x_u^{\text{out}})), \tag{8}$$

where $\mathbb{H}$ denotes the cross-entropy loss, $\lambda$ is a hyperparameter, and $q(x_u^{\text{out}})$ defines a specific assignment of distribution over $\mathcal{Y}$ for $x_u^{\text{out}}$. In this paper, we propose to assign $q(x_u^{\text{out}})$ based on the class-wise similarity scores $\text{sim}_c(x_u^{\text{out}})$ defined in (6), again utilizing the contrastive representation $f_e$ and $g$:

$$q_c(x_u) := \frac{\exp\left(\text{sim}_c(x_u; f_e, g)/\tau\right)}{\sum_i \exp\left(\text{sim}_i(x_u; f_e, g)/\tau\right)}, \tag{9}$$

where $\tau$ is a (temperature) hyperparameter.

At first glance, assigning a label of $\mathcal{Y}$ to $x_u^{\text{out}}$ may seem counter-intuitive, as the true label of $x_u^{\text{out}}$ is not in $\mathcal{Y}$ by definition. However, even when out-of-class samples cannot be represented as one-hot labels, one can still model their *class-conditional likelihoods* as a linear combination (*i.e.*, soft-label) of $\mathcal{Y}$: for instance, although "cat" images are out-of-class for CIFAR-100, still there are some classes in CIFAR-100 that is *semantically similar* to "cat", *e.g.*, "leopard", "lion", or "tiger", so that assigning a soft-label, *e.g.*, $0.1 \cdot$ "leopard" $+ 0.2 \cdot$ "lion" $+ 0.7 \cdot$ "tiger", might be beneficial. Even if out-of-classes are totally different from in-classes, one can assign the uniform labels to ignore them. We empirically found that such soft-labels based on representations learned via contrastive learning offer an effective way to utilize out-of-class samples, while they are known to significantly harm in the vanilla semi-supervised learning schemes. We present detailed discussion on our soft-label assignments in Section 4.4.

**Auxiliary batch normalization.** Finally, we suggest to handle a *data-distribution shift* originated from the class-distribution mismatch (Oliver et al., 2018), *i.e.*, $\mathcal{D}_l$ and $\mathcal{D}_u^{\text{out}}$ are drawn from the different underlying distribution. This may degrade the in-class classification performance as the auxiliary loss utilizes out-of-class samples. To handle the issue, we use additional batch normalization layers (BN; Sergey Ioffe 2015) for training samples in $\mathcal{D}_u^{\text{out}}$ to disentangle those two distributions. In our experiments, we observe such *auxiliary BNs* are beneficial when using out-of-class samples via the auxiliary loss (see Section 4.4). Auxiliary BNs also have been studied in adversarial learning literature (Xie et al., 2020): decoupling BNs improves the performance of adversarial training by handling a distribution mismatch between clean and adversarial samples. In this paper, we found that a similar strategy can improve model performance in realistic semi-supervised learning.

## 4 Experiments

In this section, we verify the effectiveness of our method over a wide range of semi-supervised learning benchmarks in the presence of various out-of-class data. The full details on experimental setups can be found in Appendix A.

**Datasets.** We perform experiments on image classification tasks for several benchmarks in the literature of semi-supervised learning (Berthelot et al., 2020; Sohn et al., 2020): CIFAR-10, CIFAR-100 (Krizhevsky et al., 2009), and ImageNet (Deng et al., 2009) datasets. Specifically, we focus on settings where each dataset is extremely label-scarce: only 4 or 25 labels per class are given during training, while the rest of the training data are assumed to be unlabeled. To configure realistic semi-supervised learning scenarios, we additionally assume that unlabeled data contain samples from an external dataset: for example, in the case of CIFAR-10, we use unlabeled samples from SVHN (Netzer et al., 2011) or TinyImageNet[3] datasets.

**Baselines.** We evaluate MixMatch (Berthelot et al., 2019), ReMixMatch (Berthelot et al., 2020), and FixMatch (Sohn et al., 2020) as baselines in our experimental setup, which are considered to be

---

[3] https://tiny-imagenet.herokuapp.com/

state-of-the-art methods in conventional semi-supervised learning. We also compare our method with three prior works applicable to our setting: namely, we consider Uncertainty-Aware Self-Distillation (UASD; Chen et al. 2020c), RealMix (Nair et al., 2019) and DS$^3$L (Guo et al., 2020), which propose schemes to detect and filter out out-of-class samples in the unlabeled dataset: *e.g.*, DS$^3$L learns to re-weight unlabeled samples to reduce the effect of out-of-class samples. Recall that our method uses SimCLR (Chen et al., 2020a) for pre-training. Unless otherwise noted, we also pre-train the baselines via SimCLR for a fair comparison, denoting those fine-tuned models by "-ft," *e.g.*, MixMatch-ft and UASD-ft. We confirm that fine-tuned models show comparable or better performance compared to those trained from scratch, as presented in Figure 1(b) and Appendix A.2. Also, we report the performance purely obtainable from (unsupervised) SimCLR: namely, we additionally consider (a) *SimCLR-le*: a SimCLR model with linear evaluation protocol (Zhang et al., 2016; Chen et al., 2020a), *i.e.*, it additionally learns a linear layer with the labeled dataset, and (b) *SimCLR-ft*: the whole SimCLR model is fine-tuned with the labeled dataset. Somewhat interestingly, these models turn out to be the strongest baselines in our setups; they often outperform the state-of-the-art semi-supervised baselines under large proportions of out-of-class samples (see Table 1). Finally, we remark that our framework can incorporate any conventional semi-supervised methods for training. We denote our method built upon an existing method by "+ OpenCoS", *e.g.*, ReMixMatch + OpenCoS.

**Training details.** As suggested by Oliver et al. (2018), we have re-implemented all baseline methods considered, including SimCLR, under the same codebase and performed experiments with the same model architecture of ResNet-50 (He et al., 2016).[4] Due to the label-scarce nature of semi-supervised learning, we do not use a validation set in our setting. Instead, we checkpoint per $2^{16}$ training samples and report (a) the median test accuracy of the last 5 checkpoints out of 50 checkpoints in total and (b) the best accuracy among all the checkpoints. We fix $\tau = 1$, the temperature hyperparameter in (9), and $\lambda = 0.5$ in (8), in all our experiments. The full details on model architecture and hyperparameters can be found in Appendix A.2 and B.1, respectively.

### 4.1 EXPERIMENTS ON VARYING PROPORTIONS OF OUT-OF-CLASS SAMPLES

We first evaluate the effect of out-of-class unlabeled samples in semi-supervised learning, on varying proportions to the total dataset. We consider CIFAR-10 and TinyImageNet datasets, and synthetically control the proportion between the two in 50K training samples. For example, 80% of proportion means the training dataset consists of 40K samples from TinyImageNet, and 10K samples from CIFAR-10. In this experiment, we assume that 25 labels per class are always given in the CIFAR-10 side. We compare three models on varying proportions of out-of-class: (a) a ReMixMatch model trained from scratch (ReMixMatch), (b) a SimCLR model fine-tuned by ReMixMatch (ReMixMatch-ft), and (c) our OpenCoS model applied to ReMixMatch-ft (+ OpenCoS).

Figure 1(b) demonstrates the results. Overall, we observe that the performance of ReMixMatch rapidly degrades as the proportion of out-of-class samples increases in unlabeled data. While ReMixMatch-ft significantly mitigates this problem, however, it still fails at a larger proportion: *e.g.*, at 80% of out-of-class, the performance of ReMixMatch-ft falls into that of ReMixMatch. OpenCoS, in contrast, successfully prevents the performance degradation of ReMixMatch-ft, especially at the regime that out-of-class samples dominate in-class samples.

### 4.2 EXPERIMENTS ON CIFAR DATASETS

In this section, we evaluate our method on several benchmarks where CIFAR datasets are assumed to be in-class: more specifically, we consider scenarios that either CIFAR-10 or CIFAR-100 is an in-class dataset, with an out-of-class dataset of either SVHN or TinyImageNet. Additionally, we also consider a separate benchmark called *CIFAR-Animals + CIFAR-Others* following the setup in the related work (Oliver et al., 2018): the in-class dataset consists of 6 animal classes from CIFAR-10, while the remaining samples are considered as out-of-class. We fix every benchmark to have 50K training samples. We assume an 80% proportion of out-of-class, *i.e.*, 10K for in-class and 40K for out-of-class samples, except for CIFAR-Animals + CIFAR-Others, which consists of 30K and 20K samples for in- and out-of-class, respectively. We report ReMixMatch-ft + OpenCoS as it tends to outperform FixMatch-ft + OpenCoS in such CIFAR-scale experiments, while FixMatch-ft + OpenCoS does in the large-scale ImageNet experiments in Section 4.3. Table 1 shows the results: OpenCoS consistently

---

[4]Note that this architecture is larger than Wide-ResNet-28-2 (Zagoruyko & Komodakis, 2016) used in the semi-supervised learning literature (Oliver et al., 2018). We use ResNet-50 following the standard of SimCLR.

Table 1: Comparison of median test accuracy on various benchmark datasets. We report the mean and standard deviation over three runs with different random seeds and splits, and also report the mean of the best accuracy in parentheses. The best scores are indicated in bold. We denote methods handling unlabeled out-of-class samples (*i.e.*, open-set) as "Open-SSL".

| In-class | | CIFAR-Animals | CIFAR-10 | | CIFAR-100 | |
|---|---|---|---|---|---|---|
| Out-of-class | Open-SSL | CIFAR-Others | SVHN | TinyImageNet | SVHN | TinyImageNet |
| *# labels per class = 4* | | | | | | |
| SimCLR-le | - | $65.58_{\pm 3.51}$ | $56.89_{\pm 3.19}$ | $58.20_{\pm 0.88}$ | $22.86_{\pm 0.17}$ | $27.93_{\pm 0.67}$ |
| SimCLR-ft | - | $67.29_{\pm 2.76 (68.25)}$ | $42.16_{\pm 2.50 (42.67)}$ | $54.26_{\pm 1.26 (55.01)}$ | $18.99_{\pm 0.04 (19.12)}$ | $29.57_{\pm 0.33 (29.57)}$ |
| UASD-ft | ✓ | $43.92_{\pm 1.94 (52.87)}$ | $42.99_{\pm 3.05 (44.70)}$ | $50.38_{\pm 2.78 (51.66)}$ | $19.66_{\pm 0.44 (19.92)}$ | $25.72_{\pm 0.69 (26.33)}$ |
| RealMix-ft | ✓ | $64.42_{\pm 7.26 (67.99)}$ | $38.22_{\pm 3.41 (41.55)}$ | $48.28_{\pm 5.73 (49.78)}$ | $18.48_{\pm 0.42 (20.04)}$ | $22.14_{\pm 0.71 (26.51)}$ |
| DS$^3$L-ft | ✓ | $63.98_{\pm 6.96 (72.20)}$ | $36.81_{\pm 7.67 (47.32)}$ | $56.32_{\pm 1.31 (57.58)}$ | $16.35_{\pm 0.20 (16.97)}$ | $23.95_{\pm 1.43 (25.06)}$ |
| MixMatch-ft | - | $44.34_{\pm 5.13 (65.55)}$ | $23.71_{\pm 8.65 (38.69)}$ | $38.90_{\pm 4.24 (46.59)}$ | $13.45_{\pm 1.23 (16.76)}$ | $23.16_{\pm 1.85 (26.54)}$ |
| FixMatch-ft | - | $34.94_{\pm 6.18 (75.83)}$ | $32.70_{\pm 6.28 (55.58)}$ | $35.99_{\pm 2.63 (63.35)}$ | $23.56_{\pm 0.68 (24.24)}$ | $30.70_{\pm 3.67 (32.52)}$ |
| ReMixMatch-ft | - | $47.61_{\pm 6.51 (64.06)}$ | $24.56_{\pm 3.99 (47.65)}$ | $28.51_{\pm 5.87 (55.68)}$ | $9.36_{\pm 1.97 (21.30)}$ | $22.33_{\pm 1.10 (29.77)}$ |
| + OpenCoS | ✓ | $\mathbf{77.66}_{\pm 3.47 (79.06)}$ | $\mathbf{61.33}_{\pm 2.88 (62.26)}$ | $\mathbf{68.37}_{\pm 5.95 (68.71)}$ | $\mathbf{28.43}_{\pm 2.42 (28.97)}$ | $\mathbf{36.51}_{\pm 1.44 (37.29)}$ |
| *# labels per class = 25* | | | | | | |
| SimCLR-le | - | $80.03_{\pm 0.73}$ | $70.31_{\pm 0.14}$ | $71.84_{\pm 0.10}$ | $37.74_{\pm 0.42}$ | $43.68_{\pm 0.26}$ |
| SimCLR-ft | - | $81.44_{\pm 0.49 (81.61)}$ | $64.41_{\pm 1.37 (64.65)}$ | $73.05_{\pm 0.11 (73.30)}$ | $39.61_{\pm 0.28 (39.87)}$ | $49.69_{\pm 0.30 (49.96)}$ |
| UASD-ft | ✓ | $82.17_{\pm 0.85 (82.50)}$ | $66.70_{\pm 1.00 (67.43)}$ | $73.97_{\pm 0.37 (74.54)}$ | $39.51_{\pm 0.76 (39.65)}$ | $44.58_{\pm 0.77 (44.90)}$ |
| RealMix-ft | ✓ | $80.27_{\pm 2.64 (81.04)}$ | $58.15_{\pm 5.27 (67.27)}$ | $69.19_{\pm 2.31 (72.29)}$ | $44.14_{\pm 1.01 (44.89)}$ | $47.57_{\pm 1.39 (49.47)}$ |
| DS$^3$L-ft | ✓ | $81.31_{\pm 0.50 (83.27)}$ | $50.00_{\pm 8.34 (63.11)}$ | $69.13_{\pm 2.30 (72.23)}$ | $29.00_{\pm 0.97 (30.17)}$ | $40.16_{\pm 0.90 (41.82)}$ |
| MixMatch-ft | - | $83.88_{\pm 1.60 (84.21)}$ | $17.98_{\pm 2.60 (54.19)}$ | $69.27_{\pm 6.59 (75.11)}$ | $38.60_{\pm 1.86 (43.02)}$ | $50.23_{\pm 0.89 (51.38)}$ |
| FixMatch-ft | - | $69.86_{\pm 1.92 (84.24)}$ | $68.02_{\pm 0.68 (71.91)}$ | $70.49_{\pm 1.15 (77.27)}$ | $41.73_{\pm 1.29 (42.28)}$ | $45.94_{\pm 1.03 (49.96)}$ |
| ReMixMatch-ft | - | $81.62_{\pm 1.47 (83.90)}$ | $37.98_{\pm 3.43 (65.33)}$ | $67.38_{\pm 7.25 (73.34)}$ | $32.75_{\pm 0.77 (44.62)}$ | $49.63_{\pm 1.10 (53.20)}$ |
| + OpenCoS | ✓ | $\mathbf{86.89}_{\pm 2.19 (87.33)}$ | $\mathbf{78.84}_{\pm 1.14 (79.25)}$ | $\mathbf{82.46}_{\pm 1.19 (82.73)}$ | $\mathbf{49.02}_{\pm 1.20 (49.53)}$ | $\mathbf{54.09}_{\pm 1.69 (54.52)}$ |

improves ReMixMatch-ft, outperforming the other baselines simultaneously. For example, OpenCoS improves the test accuracy of ReMixMatch-ft $28.51\% \rightarrow 68.37\%$ on 4 labels per class of CIFAR-10 + TinyImageNet. Also, we observe large discrepancies between the median and best accuracy of semi-supervised learning baselines, MixMatch-ft, ReMixMatch-ft, and FixMatch-ft, especially in the extreme label-scarce scenario of 4 labels per class, *i.e.*, these methods suffer from over-fitting on out-of-class samples. One can also confirm this significant over-fitting in state-of-the-art SSL methods by comparing other baselines with detection schemes, *e.g.*, USAD-ft, RealMix-ft, and DS$^3$L-ft, which show less over-fitting but with lower best accuracy.

## 4.3 EXPERIMENTS ON IMAGENET DATASETS

We also evaluate OpenCoS on ImageNet to verify its scalability to a larger and more complex dataset. We design 9 benchmarks from ImageNet dataset, similarly to Restricted ImageNet (Tsipras et al., 2019): more specifically, we define 9 super-classes of ImageNet, each of which consists of 11∼118 sub-classes. We perform our experiments on each super-class as an individual dataset. Each of the benchmarks (a super-class) contains 25 labels per sub-class, and we use the full ImageNet as an unlabeled dataset (excluding the labeled ones). In this experiment, we checkpoint per $2^{15}$ training samples and report the median test accuracy of the last 3 out of 10. We present additional experimental details, *e.g.*, configuration of the dataset, in Appendix A.3. Table 2 shows the results: OpenCoS still effectively improves the baselines, largely surpassing SimCLR-le and SimCLR-ft as well. For example, OpenCoS improves the test accuracy on Bird to 81.78% from FixMatch-ft of 78.73%, also improving SimCLR-le of 75.81% significantly. This shows the efficacy of OpenCoS in exploiting open-set unlabeled data from unknown (but related) classes or even unseen distribution of another dataset in the real-world.

## 4.4 ABLATION STUDY

We perform an ablation study to understand further how OpenCoS works. Specifically, we assess the individual effects of the components in OpenCoS and show that each of them has an orthogonal contribution to the overall improvements. We also provide a detailed evaluation of our proposed detection score (7) compared to other out-of-distribution detection methods.

Table 2: Comparison of median test accuracy on 9 super-classes of ImageNet, which are obtained by grouping semantically similar classes in ImageNet; *Dog*, *Reptile*, *Produce*, *Bird*, *Insect*, *Food*, *Primate*, *Aquatic animal*, and *Scenery*. We report the mean and standard deviation over three runs with different random seeds and splits. The best scores are indicated in bold. We denote methods handling unlabeled out-of-class samples (*i.e.*, open-set) as "Open-SSL".

| In-class | | Dog | Reptile | Produce | Bird | Insect | Food | Primate | Aquatic | Scenery |
|---|---|---|---|---|---|---|---|---|---|---|
| Number of class | | 118 | 36 | 22 | 21 | 20 | 19 | 18 | 13 | 11 |
| Out-of-class | Open-SSL | | | | | ImageNet | | | | |
| SimCLR-le | - | $43.02_{\pm0.56}$ | $51.76_{\pm0.92}$ | $64.76_{\pm0.58}$ | $75.81_{\pm1.01}$ | $59.90_{\pm0.92}$ | $56.53_{\pm0.73}$ | $53.67_{\pm0.69}$ | $68.41_{\pm1.31}$ | $64.73_{\pm0.73}$ |
| SimCLR-ft | - | $46.72_{\pm0.63}$ | $51.76_{\pm1.42}$ | $65.21_{\pm1.05}$ | $77.37_{\pm0.95}$ | $58.93_{\pm1.50}$ | $54.63_{\pm0.79}$ | $55.29_{\pm1.79}$ | $68.82_{\pm2.15}$ | $62.79_{\pm1.15}$ |
| UASD-ft | ✓ | $45.64_{\pm0.89}$ | $53.07_{\pm0.73}$ | $67.09_{\pm0.65}$ | $78.92_{\pm0.40}$ | $61.53_{\pm1.56}$ | $55.90_{\pm1.04}$ | $56.70_{\pm0.85}$ | $70.31_{\pm0.85}$ | $64.36_{\pm1.13}$ |
| RealMix-ft | ✓ | $43.55_{\pm2.36}$ | $45.70_{\pm0.16}$ | $56.06_{\pm0.65}$ | $71.94_{\pm1.21}$ | $53.33_{\pm1.78}$ | $48.25_{\pm1.30}$ | $45.89_{\pm0.84}$ | $58.10_{\pm1.34}$ | $60.79_{\pm1.36}$ |
| MixMatch-ft | - | $43.24_{\pm0.65}$ | $43.68_{\pm3.01}$ | $56.79_{\pm1.89}$ | $71.04_{\pm2.28}$ | $57.70_{\pm0.56}$ | $52.53_{\pm0.56}$ | $52.78_{\pm0.84}$ | $62.36_{\pm2.10}$ | $60.06_{\pm0.76}$ |
| ReMixMatch-ft | - | $47.47_{\pm1.47}$ | $54.39_{\pm0.78}$ | $66.88_{\pm0.38}$ | $78.95_{\pm0.67}$ | $62.30_{\pm1.32}$ | $55.48_{\pm0.76}$ | $56.63_{\pm1.81}$ | $68.67_{\pm1.24}$ | $65.58_{\pm0.86}$ |
| FixMatch-ft | - | $49.69_{\pm0.86}$ | $54.35_{\pm0.64}$ | $67.43_{\pm1.37}$ | $78.73_{\pm1.21}$ | $62.53_{\pm2.02}$ | $54.84_{\pm0.52}$ | $57.70_{\pm1.62}$ | $69.79_{\pm0.93}$ | $64.12_{\pm1.48}$ |
| **+ OpenCoS** | ✓ | $\mathbf{50.76_{\pm0.93}}$ | $\mathbf{57.19_{\pm0.84}}$ | $\mathbf{71.82_{\pm1.34}}$ | $\mathbf{81.78_{\pm0.62}}$ | $\mathbf{65.40_{\pm1.83}}$ | $\mathbf{60.53_{\pm0.11}}$ | $\mathbf{61.37_{\pm2.47}}$ | $\mathbf{73.44_{\pm1.75}}$ | $\mathbf{66.91_{\pm1.61}}$ |

Table 3: Ablation study on three main components of our method: the detection criterion ("Detect"), auxiliary loss ("Aux. loss"), and auxiliary BNs ("Aux. BNs"). We report the mean and standard deviation over three runs with different random seeds and a fixed split of labeled data.

| In-class + Out-of-class | | | CIFAR-Animals | CIFAR-10 | Produce | Bird | Food |
|---|---|---|---|---|---|---|---|
| Detection | Aux. loss | Aux. BNs | + CIFAR-Others | + SVHN | + ImageNet | + ImageNet | + ImageNet |
| ✓ | - | - | $79.03_{\pm0.25}$ | $52.24_{\pm2.06}$ | $69.79_{\pm0.05}$ | $79.02_{\pm0.20}$ | $54.98_{\pm0.22}$ |
| ✓ | ✓ | - | $79.68_{\pm0.33}$ | $55.78_{\pm0.35}$ | $71.85_{\pm0.43}$ | $80.73_{\pm0.31}$ | $58.77_{\pm0.12}$ |
| ✓ | ✓ | ✓ | $80.02_{\pm0.71}$ | $57.77_{\pm0.76}$ | $72.48_{\pm0.19}$ | $82.32_{\pm0.05}$ | $60.00_{\pm0.21}$ |

**Component analysis.** To further analyze the individual contribution of each component of Open-CoS, we incrementally apply these components one-by-one to ReMixMatch-ft (CIFAR-scale) and FixMatch-ft (ImageNet-scale) baselines. Specifically, we consider CIFAR-Animals + CIFAR-Others, CIFAR-10 + SVHN for CIFAR-scale, and Produce, Bird, Food + ImageNet for ImageNet-scale benchmarks. Table 3 summarizes the results, and they indeed confirm that each of what comprises OpenCoS has an orthogonal contribution to improve the accuracy of the benchmarks tested. We observe that leveraging out-of-class samples via auxiliary loss ("Aux. loss") achieves consistent improvements, and also outperforms the baselines significantly. Finally, we remark auxiliary batch normalization layers ("Aux. BNs") give a consistent improvement, and it is often significant: *e.g.*, it gives 55.78% → 57.77% on CIFAR-10 + SVHN.

**Other detection scores.** In Section 3.2, we propose a detection score $s(\cdot)$ (7) for detecting out-of-class samples in an unlabeled dataset, based on the contrastive representation of SimCLR. This setup is different to the standard *out-of-distribution* (OOD) detection task (Emmott et al., 2013; Liu et al., 2018): OOD detection targets *unseen* (*i.e.*, "out-of-distribution") samples in test time, while our setup aims to detect *seen* out-of-class samples during training assuming few in-class labels. Due to this lack of labeled information, therefore, the standard techniques for OOD detection (Hendrycks & Gimpel, 2017; Liang et al., 2018; Lee et al., 2018b) are not guaranteed to perform still well in our setup. We examine this in Appendix B.3 by comparing detection performance of such OOD detection scores with ours (7) upon a shared SimCLR representation: in short, we indeed observe that our approach of directly leveraging the contrastive representation could perform better than simply applying OOD scores relying on few labeled samples, *e.g.*, our score achieves an AUROC of 98.10% on the CIFAR-Animals + CIFAR-Others benchmark compared to the maximum softmax probability based score (Hendrycks & Gimpel, 2017) of 80.79%. We present the detailed experimental setups and more results in Appendix B.3.

**Effect of soft-labeling.** We emphasize that our soft-labeling scheme can be rather viewed as a more reasonable way to label such out-of-class samples compared to existing state-of-the-art SSL methods, *e.g.*, MixMatch simply assigns its sharpened predictions. On the other hand, a prior work (Li & Hoiem, 2016) has a similar observation to our approach: assigning soft-labels of novel data could be beneficial for transfer learning. This motivate us to consider an experiment to further support the claim that our soft-labeling gives informative signals: we train a classifier by minimizing only the cross-entropy loss with soft-labels (*i.e.*, without in-class samples) *from scratch*. In Table 4, the trained

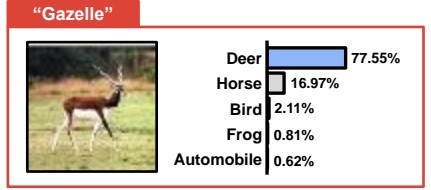
(a) Top-5 classes in a soft-label of "Gazelle".

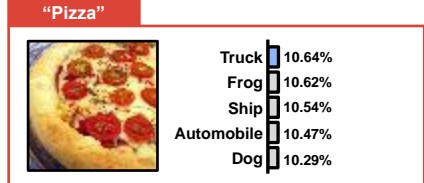
(b) Top-5 classes in a soft-label of "Pizza".

Figure 3: Illustration of soft-label assignments in the CIFAR-10 + TinyImageNet benchmark. Un-labeled out-of-class samples from (a) "gazelle" is assigned with soft-labels of ≈78% confidence for "deer", and (b) "pizza" is assigned with almost uniform soft-labels (≈10% of confidence). The soft-labels are scaled with the temperature $\tau = 0.1$.

Table 4: Comparison of the median test accuracy of ResNet-50 trained on out-of-class samples and their soft-labels of the CIFAR-10 benchmarks with 4 labels per class. We denote the new setting of minimizing with the auxiliary loss as "Aux. loss only". We report the mean and standard deviation over three runs with different random seeds and splits of labeled data.

| In-class | CIFAR-Animals | CIFAR-10 | |
|---|---|---|---|
| Out-of-class | + CIFAR-Others | + SVHN | + TinyImageNet |
| SimCLR-le | $65.58_{\pm 3.51}$ | $56.89_{\pm 3.19}$ | $58.20_{\pm 0.88}$ |
| ReMixMatch-ft | $47.61_{\pm 6.51}$ | $24.56_{\pm 3.99}$ | $28.51_{\pm 5.87}$ |
| Aux. loss only | $\mathbf{39.76_{\pm 1.97}}$ | $\mathbf{25.97_{\pm 3.92}}$ | $\mathbf{21.19_{\pm 5.49}}$ |

classifier performs much better than (random) guessing, even close to some baselines although this model does; this supports that generated soft-labels contain informative features of in-classes. The details on experimental setups can be found in Appendix A.2.

**Examples of actual soft-labels.** We also present some concrete examples of our soft-labeling scheme in Figure 3 for a better understanding, which are obtained from random unlabeled samples in the CIFAR-10 + TinyImageNet benchmark: Overall, we qualitatively observe that out-of-class samples that share some semantic features to the in-classes (*e.g.*, Figure 3(a)) have relatively high con-fidence capturing such similarity, while returning very close to uniform otherwise (*e.g.*, Figure 3(b)).

## 5 CONCLUSION

In this paper, we propose a simple and general framework for handling novel unlabeled data, aiming toward a more realistic assumption for semi-supervised learning. Our key idea is (intentionally) not to use label information, *i.e.*, by relying on *unsupervised* representation, when handling novel data, which can be naturally incorporated into semi-supervised learning with our framework: OpenCoS. In contrast to previous approaches, OpenCoS opens a way to further utilize those novel data by assigning them soft-labels, which are again obtained from unsupervised learning. We hope our work would motivate researchers to extend this framework with a more realistic assumption, *e.g.*, noisy labels (Wang et al., 2018; Lee et al., 2019), imbalanced learning (Liu et al., 2020).

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

# A    Training details

## A.1    Details on the experimental setup

For the experiments reported in Table 1, we generally follow the training details of FixMatch (Sohn et al., 2020), including optimizer, learning rate schedule, and an exponential moving average. Specifically, we use Nesterov SGD optimizer with momentum 0.9, a cosine learning rate decay with an initial learning rate of 0.03, and an exponential moving average with a decay of 0.999. The batch size is 64, which is widely adopted in semi-supervised learning (SSL) methods. We do not use weight decay for these models, as they are fine-tuned. We use a simple augmentation strategy, *i.e.*, flip and crop, as a default. We use the augmentation scheme of SimCLR (Chen et al., 2020a) (*i.e.*, random crop with resize, random color distortion, and random Gaussian blur) when a SSL method requires to specify a *strong* augmentation strategy, *e.g.*, for consistency regularization in the SSL literature (Berthelot et al., 2020; Sohn et al., 2020). We fix the number of augmentations as 2, following Berthelot et al. (2019): *e.g.*, MixMatch-ft generates two augmentations of each unlabeled sample while ReMixMatch-ft generates one weak augmentation and one strong augmentation. In the case of ReMixMatch-ft, we do not use the ramp-up weighting function, the pre-mixup and the rotation loss, which give a marginal difference in fine-tuning, for efficient computation.[5] For FixMatch-ft, we set the relative size of labeled and unlabeled batch $\mu = 1$ for a fair comparison with other baselines, and scale the learning rate linearly with $\mu$, as suggested by Sohn et al. (2020). Following Chen et al. (2020c), UASD-ft computes the predictions by accumulative ensembling instead of using an exponential moving average. OpenCoS shares all hyperparameters of the baseline SSL methods, *e.g.*, FixMatch + OpenCoS shares hyperparameters of FixMatch-ft. For the results of ReMixMatch (from scratch) in Figure 1(b), we report the median accuracy of the last 10 checkpoints out of 200 checkpoints, where each checkpoint is saved per $2^{16}$ training samples.

## A.2    CIFAR experiments

**Training from scratch.**  We pre-train all the baselines via SimCLR for a fair comparison, as mentioned in Section 4. In Table 5, we also report the performance of each baseline model when trained from scratch. Here, we report the median accuracy of the last 10 checkpoints out of 500 checkpoints in total. We also present the fine-tuned baselines (see Section 4.2) denoting by "-ft," *e.g.*, MixMatch-ft. Here, we follow the training details those which originally used in each baseline method. For example, ReMixMatch from scratch uses Adam optimizer with a fixed learning rate of 0.002, and weight decay of 0.02. We use the simple strategy (*i.e.*, flip and crop) and RandAugment (Cubuk et al., 2019) as a weak and strong augmentation, respectively. In addition, we use the ramp-up weighting function, the pre-mixup and the rotation loss for ReMixMatch. We consider the CIFAR-100 + TinyImageNet benchmark assuming 80% proportion of out-of-class, *i.e.*, 10K samples for in-class and 40K samples for out-of-class.

**Training without in-class samples from scratch.**  For the experiments reported in Table 4, we train a classifier from scratch only with the unlabeled out-of-class samples and their soft-labels, on the CIFAR-10 benchmarks with 4 labels per class. We use ResNet-50, and SGD with momentum 0.9, weight decay 0.0001, and an initial learning rate of 0.1. The learning rate is divided by 10 after epochs 100 and 150, and total epochs are 200. We set batch size as 128, and use a simple data augmentation strategy, *i.e.*, flip and crop. We minimize the cross-entropy loss between the soft-labels $q(\cdot)$ and model predictions $f(\cdot)$, *i.e.*, $\mathcal{L} = \mathbb{H}(q(x_u^{\text{out}}), f(x_u^{\text{out}}))$. We use the temperature scaling on the both sides of soft-labels and model predictions for a stable training, specifically 0.1 and 4, respectively.

**Analysis of model architectures.** For all our experiments, we use ResNet-50 following the standard of SimCLR (Chen et al., 2020a). This architecture is larger than Wide-ResNet-28-2 (Zagoruyko & Komodakis, 2016), a more widely adopted architecture in the semi-supervised learning literature (Oliver et al., 2018). We have found that using a larger network, *i.e.*, ResNet-50, is necessary to leverage the pre-trained features of SimCLR: In Table 5, we provide an evaluation on another choice of model architecture, *i.e.*, Wide-ResNet-28-2. The hyperparameters are the same as the experiments on ResNet-50. Here, one can observe that OpenCoS trained on Wide-ResNet-28-2 still improves ReMixMatch-ft, outperforming the other baselines. More importantly, however, we observe that pre-training Wide-ResNet-28-2 via SimCLR does not significantly improve the baselines trained from

---

[5]For a fair comparison, ReMixMatch-ft + OpenCoS shares these settings.

Table 5: Comparison of the median test accuracy of Wide-ResNet-28-2 and ResNet-50 on CIFAR-100 + TinyImageNet benchmark over baseline methods. The best scores are indicated in bold. We denote methods handling unlabeled out-of-class samples (*i.e.*, open-set) as "Open-SSL".

| In-class | | CIFAR-100 | | | |
|---|---|---|---|---|---|
| Out-of-class | | TinyImageNet | | | |
| Model architecture | | Wide-ResNet-28-2 | | ResNet-50 | |
| Labels per class | Open-SSL | 4 | 25 | 4 | 25 |
| SimCLR-le | - | 18.14 | 33.48 | 27.93 | 43.68 |
| SimCLR-ft | - | 17.55 | 36.94 | 29.57 | 49.69 |
| UASD | ✓ | 8.76 | 27.62 | 7.80 | 19.21 |
| UASD-ft | ✓ | 11.48 | 27.65 | 25.72 | 44.58 |
| RealMix | ✓ | 13.15 | 36.97 | 12.15 | 28.46 |
| RealMix-ft | ✓ | 13.56 | 33.31 | 22.14 | 47.57 |
| MixMatch | - | 14.83 | 37.94 | 14.19 | 32.49 |
| MixMatch-ft | - | 13.11 | 39.66 | 23.16 | 50.23 |
| FixMatch | - | 20.68 | 44.33 | 27.75 | 44.78 |
| FixMatch-ft | - | 16.59 | 36.84 | 30.70 | 45.94 |
| ReMixMatch | - | 16.06 | 40.45 | 17.87 | 39.86 |
| ReMixMatch-ft | - | 16.94 | 45.21 | 22.33 | 49.63 |
| + OpenCoS (ours) | ✓ | **27.45** | **46.95** | **36.51** | **54.09** |

scratch, contrary to the results of ResNet-50. As also explored by Chen et al. (2020a), we suspect this is due to that pre-training via SimCLR requires a larger model in practice, and suggest future SSL research to explore larger architectures to incorporate more rich features into their methods, *e.g.*, features learned via unsupervised learning (Hénaff et al., 2019; Chen et al., 2020a;b).

**Experiments on more labeled data.** We have performed additional experiments on the CIFAR-10 + SVHN benchmark with 400 labels per class, and the results are given in Table 6. One can still observe that OpenCoS consistently outperforms other methods when more labeled data are available.

Table 6: Comparison of the median test accuracy on the CIFAR-10 + SVHN benchmark with 400 labels per class over baseline methods. The best scores are indicated in bold.

| | SimCLR-le | SimCLR-ft | UASD-ft | RealMix-ft | MixMatch-ft | FixMatch-ft | ReMixMatch-ft | + OpenCoS (ours) |
|---|---|---|---|---|---|---|---|---|
| Accuracy | 78.13 | 82.44 | 82.48 | 85.76 | 77.10 | 83.91 | 83.60 | **88.38** |

## A.3 IMAGENET EXPERIMENTS

**Benchmarks of ImageNet dataset.** In Section 4.3, we introduce 9 benchmarks from ImageNet dataset, similar to Restricted ImageNet (Tsipras et al., 2019). In detail, we group together subsets of semantically similar classes into 9 different super-classes, as shown in Table 7.

**Details of ImageNet experiments.** For the experiments reported in Table 2, we use a pre-trained ResNet-50 model[6] of Chen et al. (2020a) and fine-tune the projection header for 5 epochs on ImageNet. We follow the optimization details of the fine-tuning experiments of SimCLR (Chen et al., 2020a): specifically, we use Nesterov SGD optimizer with momentum 0.9, and a learning rate of 0.00625 (following LearningRate = $0.05 \cdot$ BatchSize$/256$). We set the batch size to 32, and report the median accuracy of the last 3 checkpoints out of 10 checkpoints in total. Data augmentation, regularization techniques, and other hyperparameters are the same as CIFAR experiments. In the case of FixMatch-ft + OpenCoS, we empirically observe that it is more beneficial not to discard the detected out-of-class samples in FixMatch training, as it performs better than using in-class samples

---

[6]https://github.com/google-research/simclr

Table 7: Super-classes used in 9 benchmarks from ImageNet dataset. The class ranges are inclusive.

| Super-class | Corresponding ImageNet Classes |
|---|---|
| "Dog" | 151 to 268 |
| "Reptile" | 33 to 68 |
| "Produce" | 936 to 957 |
| "Bird" | 80 to 100 |
| "Insect" | 300 to 319 |
| "Food" | 928 to 935 & 959 to 969 |
| "Primate" | 365 to 382 |
| "Aquatic" | 118 to 121 & 389 to 397 |
| "Scenery" | 970 to 980 |

only: the auxiliary loss still use only out-of-class samples. Since FixMatch filters low-confidence unlabeled samples out, it is possibly due to a decrease in the number of training data.

## B  ABLATION STUDY

### B.1  EFFECTS OF THE TEMPERATURE AND THE LOSS WEIGHT

In Section 4, we perform all the experiments with the fixed temperature $\tau = 1$ and loss weight $\lambda = 0.5$. To examine the effect of hyperparameters $\tau$ and $\lambda$, we additionally test the hyperparameters across an array of $\tau \in \{0.1, 0.5, 1, 2, 4\}$ and $\lambda \in \{0.1, 0.5, 1, 2, 4\}$ on the CIFAR-100 + TinyImageNet benchmark with ResNet-50. The results are presented in Table 8. Overall, we found our method is fairly robust on $\tau$ and $\lambda$.

Table 8: Comparison of median test accuracy on the CIFAR-100 + TinyImageNet benchmark with (a) 4 and (b) 25 labels per class, over various hyperparameters $\tau$ and $\lambda$.

| (a) 4 labels per class | | | | | | (b) 25 labels per class | | | | | |
|---|---|---|---|---|---|---|---|---|---|---|---|
| $\tau$ \ $\lambda$ | 0.1 | 0.5 | 1 | 2 | 4 | $\tau$ \ $\lambda$ | 0.1 | 0.5 | 1 | 2 | 4 |
| 0.1 | 37.07 | 37.62 | 36.89 | 37.63 | 37.76 | 0.1 | 55.86 | 55.71 | 55.99 | 55.64 | 55.44 |
| 0.5 | 38.22 | 38.02 | 36.48 | 37.51 | 38.18 | 0.5 | 54.70 | 56.14 | 56.30 | 55.62 | 56.09 |
| 1 | 37.76 | 37.46 | 37.54 | 37.45 | 37.31 | 1 | 55.66 | 56.01 | 56.12 | 55.45 | 55.99 |
| 2 | 37.06 | 37.86 | 36.92 | 37.15 | 37.12 | 2 | 55.67 | 55.79 | 55.95 | 55.96 | 56.04 |
| 4 | 37.14 | 37.41 | 37.94 | 36.94 | 37.43 | 4 | 55.75 | 55.58 | 56.23 | 56.56 | 55.31 |

### B.2  EFFECTS OF OUT-OF-CLASS SAMPLES IN CONTRASTIVE LEARNING

To clarify how the improvements of OpenCoS comes from out-of-class samples, we have considered additional CIFAR-scale experiments with 4 labels per class. We newly pre-train and fine-tune SimCLR models using in-class samples only, *i.e.*, 30,000 for CIFAR-Animals, 10,000 for CIFAR-10, CIFAR-100 benchmarks, and compare two baselines: SimCLR-le and ReMixMatch-ft. Interestingly, we found that just merging out-of-class samples to the training dataset improves the performance of SimCLR models in several cases (see Table 9), *e.g.*, SimCLR-le of CIFAR-10 enhances from 55.27% to 58.20% with TinyImageNet. Also, OpenCoS significantly outperforms overall baselines, even when out-of-class samples hurt the performance of SimCLR-le or ReMixMatch-ft. We confirm that the proposed method effectively utilizes contrastive representations of out-of-class samples beneficially, compared to other SSL baselines.

**Robustness to incorrect detection.** We observe our method is quite robust on incorrectly detected out-of-class samples, *i.e.*, those samples are still leveraged via auxiliary loss instead of SSL algorithm. We have considered an additional experiment on CIFAR-10 with 250 labels (out of 50,000 samples), that assumes (*i*) all the unlabeled samples are in-class, and (*ii*) 80% of those in-class samples are

Table 9: Comparison of the median test accuracy for the use of out-of-class samples on the CIFAR-scale benchmarks with 4 labels per class. We denote whether using out-of-class samples for training as "w/ out-of-class". We report the mean and standard deviation over three runs with different random seeds and splits. The best scores are indicated in bold.

| In-class + Out-of-class | | CIFAR-Animals | CIFAR-10 | | CIFAR-100 | |
|---|---|---|---|---|---|---|
| Method | w/ out-of-class | + CIFAR-Others | + SVHN | + TinyImageNet | + SVHN | + TinyImageNet |
| SimCLR-le | - | $64.30_{\pm 6.50}$ | $55.27_{\pm 2.99}$ | | $25.17_{\pm 1.08}$ | |
| SimCLR-le | ✓ | $65.58_{\pm 3.51}$ | $56.89_{\pm 3.19}$ | $58.20_{\pm 0.88}$ | $22.86_{\pm 0.17}$ | $27.93_{\pm 0.67}$ |
| ReMixMatch-ft | - | $54.04_{\pm 7.05}$ | $28.15_{\pm 0.72}$ | | $16.55_{\pm 3.04}$ | |
| ReMixMatch-ft | ✓ | $47.61_{\pm 6.51}$ | $24.56_{\pm 3.99}$ | $28.51_{\pm 5.87}$ | $9.36_{\pm 1.97}$ | $22.33_{\pm 1.10}$ |
| **+ OpenCoS (ours)** | ✓ | $\mathbf{77.66_{\pm 3.47}}$ | $\mathbf{61.33_{\pm 2.88}}$ | $\mathbf{68.37_{\pm 5.95}}$ | $\mathbf{28.43_{\pm 2.42}}$ | $\mathbf{36.51_{\pm 1.44}}$ |

incorrectly detected as out-of-class in OpenCoS. Here, we compare OpenCoS with a baseline which only uses the correctly-detected (in-class) samples without auxiliary loss, *i.e.*, the baseline is trained on 10,000 samples while OpenCoS on 50,000 in total. In this scenario, OpenCoS achieves 89.54% in the median test accuracy, while the baseline does 89.27%: this shows that our auxiliary loss does not harm the training even when it is incorrectly applied to in-class samples.

## B.3 EVALUATIONS OF OUR DETECTION SCORE

**Baselines.** We consider maximum softmax probability (MSP; Hendrycks & Gimpel 2017), ODIN (Liang et al., 2018), and Mahalanobis distance-based score (Lee et al., 2018b) as baseline detection methods. As MSP and ODIN require a classifier to obtain their scores, we employ SimCLR-le: a SimCLR model which additionally learns a linear layer with the labeled dataset, for both baselines.

ODIN performs an input pre-processing by adding small perturbations with a temperature scaling as follows:

$$P(y = c|x; T) = \frac{\exp(f_c(x)/T)}{\sum_y \exp(f_y(x)/T)}, \quad x' = x - \epsilon \cdot \text{sign}(-\nabla_x \log P(y = c|x; T)), \quad (10)$$

where $f = (f_1, ..., f_C)$ is the logit vector of deep neural network, $T > 0$ is a temperature scaling parameter, and $\epsilon$ is a magnitude of noise. ODIN calculates the pre-processed data $x'$ and feeds it into the classifier to compute the confidence score, *i.e.*, $\max_y P(y|x'; T)$, and identifies it as in-class if the confidence score is higher than some threshold $\delta$. We choose the temperature $T$ and the noise magnitude $\epsilon$ from $\{1, 10, 100, 1000\}$ and $\{0, 0.0005, 0.001, 0.0014, 0.002, 0.0024, 0.005, 0.01, 0.05, 0.1, 0.2\}$, respectively, by using 2,000 validation data.

Mahalanobis distance-based score (Mahalanobis) assumes the features of the neural network $f$ follows the class-conditional Gaussian distribution. Then, it computes Mahalanobis distance between input $x$ and the closest class-conditional Gaussian distribution, *i.e.*,

$$M(x) = \max_c -(f(x) - \mu_c)^\top \Sigma^{-1}(f(x) - \mu_c), \quad (11)$$

where $\mu_c$ is the class mean and $\Sigma$ is the covariance of the labeled data. We fix the covariance matrix as the identity because the number of labeled samples is insufficient to compute it: the feature dimension of SimCLR encoder $f_e$ and projection header $g$ are 2048. Moreover, Mahalanobis has the noise magnitude parameter $\epsilon$ for input pre-processing like ODIN, and use a feature ensemble method of Lee et al. (2018b). We choose $\epsilon$ from $\{0, 0.0005, 0.001, 0.0014, 0.002, 0.005, 0.01\}$, and perform the feature ensemble of intermediate features including $f_e$'s and $g$'s by using 2,000 validation data.

**Metrics.** We follow the threshold-free detection metrics used in Lee et al. (2018b) to measure the effectiveness of detection scores in identifying out-of-class samples.

- **True negative rate (TNR) at 95% true positive rate (TPR).** We denote true positive, true negative, false positive, and false negative as TP, TN, FP, and FN, respectively. We measure TNR = TN / (FP+TN) at TPR = TP / (TP+FN) is 95%.

- **Detection accuracy.** For an unlabeled data $x \in \mathcal{D}_u (= \mathcal{D}_u^{\text{in}} \cup \mathcal{D}_u^{\text{out}})$, this metric corresponds to the maximum classification probability over all possible thresholds $\delta$:

$$1 - \min_{\delta} \{ \text{FNR} \cdot P(x \in \mathcal{D}_u^{\text{in}}) + \text{FPR} \cdot P(x \in \mathcal{D}_u^{\text{out}}) \}$$

  where false negative rate FNR = FN / (FN+TP), and false positive rate FPR = FP / (FP+TN).

- **Area under the receiver operating characteristic curve (AUROC).** The receiver operating characteristic (ROC) curve is a graph of the true positive rate (TPR) against the false positive rate (FPR) by varying a threshold, and we measure its area.

- **Area under the precision-recall curve (AUPR).** The precision-recall (PR) curve is a graph of the precision = TP / (TP+FP) against recall = TP / (TP+FN) by varying a threshold. AUPR-in (or -out) is AUPR where in- (or out-of-) class samples are specified as positive.

Table 10: Comparison of detection methods on the CIFAR-Animals + CIFAR-Others benchmark with 4 labels per class under various evaluation metrics. We denote our detection method without the projection header as "Ours w/o header". The best scores are indicated in bold.

| Detection method | TNR at TPR 95% ↑ | Detection Accuracy ↑ | AUROC ↑ | AUPR-in ↑ | AUPR-out ↑ |
|---|---|---|---|---|---|
| MSP | 18.20 | 72.81 | 80.79 | 88.72 | 66.38 |
| ODIN | 24.10 | 79.41 | 86.55 | 92.55 | 72.75 |
| Mahalanobis | 90.90 | **93.57** | 97.48 | 98.60 | 94.36 |
| Ours w/o header | 40.44 | 80.11 | 88.80 | 93.46 | 79.35 |
| Ours | **91.20** | 93.49 | **98.10** | **98.79** | **96.86** |

In this section, we present evaluations of our detection score $s(\cdot)$ (7) under various detection metrics on the CIFAR-Animals + CIFAR-Others with 4 labels per class. Table 10 shows the results: interestingly, our score outperforms MSP and ODIN and also performs comparable to Mahalanobis, even these baselines require more computational costs, *e.g.*, input pre-processing. We confirm that the design of our score is an effective and efficient way to detect out-of-class samples based on the representation of SimCLR. In Figure 4, we provide the receiver operating characteristic (ROC) curves that support the above results. We remark that the projection header $g$ (4) is crucial for the detection, *e.g.*, $g$ enhances AUROC of our score from 88.80% to 98.10%. According to the definition of our score (7), it can be viewed as a simpler version of Mahalanobis without its input pre-processing and feature ensembles under an assumption of identity covariance, which may explain their comparable performances.

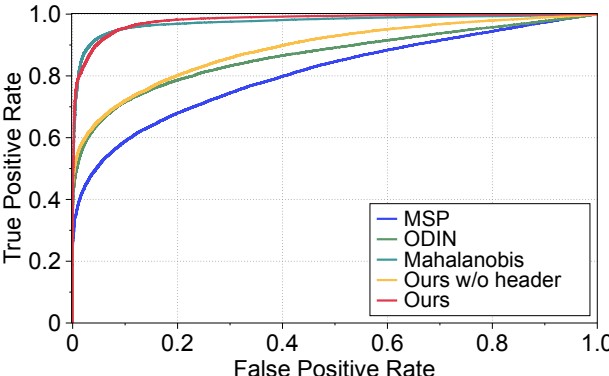

Figure 4: Receiver operating characteristic (ROC) curves of detection methods on CIFAR-Animals + CIFAR-Others benchmark with 4 labels per class.

We additionally provide the performance of OpenCoS among various detection methods, including the above baselines and two artificial methods: we consider (a) *Random*: a random detection with a probability of 0.5, and (b) *Oracle*: a perfect detection. For MSP, ODIN, and Mahalanobis, we choose their detection thresholds at TPR 95%. Table 11 shows the results: we observe that the classification accuracy is proportional to the detection performance. Remarkably, our detection method achieves comparable accuracy to Oracle, which is the optimal performance of OpenCoS.

Table 11: Comparison of the median test accuracy on the CIFAR-Animals + CIFAR-Others benchmark with 4 labels per class among various detection methods. We denote our detection method without the projection header as "Ours w/o header". We report mean and standard deviation over three runs with different random seeds and a fixed split of labeled data. The best scores are indicated in bold.

| Detection method | Random | MSP | ODIN | Mahalanobis | Ours w/o header | Ours | Oracle |
|---|---|---|---|---|---|---|---|
| Accuracy | $59.74_{\pm 3.01}$ | $61.78_{\pm 5.51}$ | $64.08_{\pm 2.00}$ | $78.03_{\pm 0.95}$ | $75.27_{\pm 0.72}$ | $80.02_{\pm 0.71}$ | $\mathbf{81.10_{\pm 1.45}}$ |

### B.4 Evaluations of the detection threshold

We additionally provide the detection performance on various proportions of out-of-class samples, *i.e.*, 50% and 67%, on this benchmark. For each setting, the number of out-of-class samples is fixed at 20K, while in-class samples are controlled to 20K and 10K, respectively. We choose the same detection threshold $t := \mu_l - 2\sigma_l$ throughout all experiments: it is a reasonable choice, giving $\approx 95\%$ confidence if the score follows Gaussian distribution. Table 12(a) shows the detection performance of our threshold and its applicability over various proportions of out-of-class samples. Although tuning $t$ gives further improvements though (see Table 12(b)), we fix the threshold without any tuning.

Table 12: The detection performance across different (a) proportions of out-of-class and (b) detection thresholds in CIFAR-Animals + CIFAR-Others benchmark with 4 labels per class.

(a) The detection performance of the proposed threshold $t := \mu_l - 2\sigma_l$ on varying proportions of out-of-class.

| Proportion of out-of-class | 40% | 50% | 67% |
|---|---|---|---|
| True Positive Rate (TPR) | 63.61 | 72.19 | 81.10 |
| True Negative Rate (TNR) | 99.76 | 99.66 | 99.55 |
| AUROC | 98.10 | 98.50 | 98.90 |

(b) The detection performance and median test accuracy across different thresholds $t$, *i.e.*, $k = 1, 2, 3, 4$ of $t = \mu_l - k \cdot \sigma_l$.

| $k$ | 1 | 2 | 3 | 4 |
|---|---|---|---|---|
| True Positive Rate (TPR) | 37.48 | 63.61 | 85.55 | 98.62 |
| True Negative Rate (TNR) | 99.94 | 99.76 | 97.22 | 74.48 |
| Accuracy | 75.47 | 80.67 | 81.12 | 78.37 |

## C  Algorithm

The full training procedure of OpenCoS is summarized in Algorithm 1.

---

**Algorithm 1** OpenCoS: A general framework for open-set semi-supervised learning (SSL).

---

1: **Input:** The classifier $f$, the encoder $f_e$ (*i.e.*, penultimate features of $f$), and the projection header $g$. A labeled dataset $\mathcal{D}_l$ and an open-set unlabeled data $\mathcal{D}_u$.

---

2: Pre-train $f_e$ and $g$ via SimCLR.
3: $S_l = \phi$
4: **for** each labeled sample $x_l \in \mathcal{D}_l$ **do**
5:      $S_l = S_l \cup \{s(x_l; f_e, g)\}$                  ▷ The similarity score (7).
6: **end for**
7: $t := \mathbb{E}\left[S_l\right] - 2\sqrt{\mathrm{Var}\left[S_l\right]}$               ▷ Compute the threshold $t$.
8:
9: $\mathcal{D}_u^{\mathrm{in}} = \phi, \mathcal{D}_u^{\mathrm{out}} = \phi$
10: **for** each unlabeled sample $x_u \in \mathcal{D}_u$ **do**
11:      **if** $s(x_u; f_e, g) < t$ **then**
12:          $\mathcal{D}_u^{\mathrm{out}} = \mathcal{D}_u^{\mathrm{out}} \cup \{x_u\}$            ▷ Detect out-of-class unlabeled samples.
13:      **else**
14:          $\mathcal{D}_u^{\mathrm{in}} = \mathcal{D}_u^{\mathrm{in}} \cup \{x_u\}$
15:      **end if**
16: **end for**
17:
18: **for** each sample $x_l \in \mathcal{D}_l$, $x_u^{\mathrm{in}} \in \mathcal{D}_u^{\mathrm{in}}$, and $x_u^{\mathrm{out}} \in \mathcal{D}_u^{\mathrm{out}}$ **do**
19:      $q(x_u^{\mathrm{out}}) \leftarrow$ Compute a soft-label of $x_u^{\mathrm{out}}$      ▷ Using contrastive representations (9).
20:      $\mathcal{L}_{\mathrm{OpenCoS}} = \mathcal{L}_{\mathrm{SSL}}(x_l, x_u^{\mathrm{in}}; f) + \lambda \cdot \mathbb{H}(q(x_u^{\mathrm{out}}), f(x_u^{\mathrm{out}}))$    ▷ SSL with auxiliary loss (8).
21:      Update parameters of $f$ by computing the gradients of the proposed loss $\mathcal{L}_{\mathrm{OpenCoS}}$.
22: **end for**

---

