# OpenReview forum: "OpenCoS: Contrastive Semi-supervised Learning for Handling Open-set Unlabeled Data"
_ICLR.cc/2021/Conference — Reject_

### Official Review · AnonReviewer2 · 2020-10-22
**The paper uses contrastive learning idea proposed in SimCLR (Chen et al. 2020) to detect out-of-class samples and treat them in a different way than in-class unlabeled samples during semi-supervised learning.**

**Rating:** 4
**Confidence:** 4

**Review:**

Quality:
The quality of the paper is below average. The loss function that integrates out-of-class samples is counter intuitive and seems to be chosen based on improved empirical evidence.

Clarity:
The paper reads well.

Originality/Significance:
The originality of the approach is limited. Main ideas are borrowed from SimCLR and applied to SSL.

Detailed Comments:

The paper uses contrastive learning idea proposed in SimCLR (Chen et al. 2020) to detect out-of-class samples and treat them in a different way than in-class unlabeled samples during semi-supervised learning. It also explores the idea of auxiliary batch normalization (from Xie et al. 2020) in the open-set SSL setting but the results of the ablation study suggest the level of improvement achieved by this normalization is negligible and the most of the improvement comes from more accurate detection of out-of-class samples through using the projection header function introduced in SimCLR paper. Although results across multiple benchmark datasets report significant improvement over other SSL techniques these improvements could be artificial as other techniques have no way of handling out-of-class samples. Integrating these other out-of-class detection techniques in ReMixMatch and comparing results with the proposed technique would offer a more compelling argument. Overall, the paper proposes a nice practical idea but for publication in ICLR one would like to see more theoretical insight along with empirical evidence.

Class conditional likelihoods has been shown to be not very useful in detecting out-of-distribution samples in cross-entropy loss learning. What aspect of contrastive learning makes it more useful for open-set classification?

Cross-entropy term in equation 8 does not make much sense. It is simply computing the loss with respect to an incorrect label. If a sample belongs to an unknown class it is not clear why this would help SSL. If the purpose here is to capture shared characteristics of the samples then SimCLR trained with both labeled and unlabeled data already takes care of it. The authors try to justify this by considering that some seen classes would be similar to the unseen one but for a fine-grained classification task such samples may still hurt predictive performance. If there are no similar classes among labeled classes the authors argue that this loss will have a uniform affect for all classes. Again, this is not a compelling argument. No matter how dissimilar the unseen sample to labeled classes are it would be more similar to some of the classes than some others. Due to the normalization affect the weight distributions will significantly deviate from a uniform distribution.

Baseline techniques are all SSL techniques. These are guaranteed to perform worse than the proposed technique because they have no way of handling out-of-class samples. What about other baselines? From Table 3 one can see that the main contribution comes from the detection of out-of-class samples. The authors compare the detection performance of their technique against other standard outlier detection techniques and show that the proposed detection outperforms all of them to achieve the highest AUC in the "isolated" detection task. However, it is still not clear why the projection header g achieves something that softmax probabilities cannot do. In other words why do class-conditional probabilities obtained from g are useful for out-of-class detection but the probabilities that one would obtain from the softmax layer of an architecture (such resnet) trained by cross-entropy loss is not much useful for the same task. Interesting  that not much insight has been provided in the SimCLR paper either. Along the same lines, as a contrastive loss function, triplet loss also seems to do well in open-world settings.

Minor:

Please correct the following:

page1 Compared to prior approaches approaches have that bypassed ...

---

> ### Author Response · Authors · 2020-11-18
> **Response to R2 (1/2)**
>
> Dear R2,
>
> Thank you for your detailed feedback to help us improve the manuscript. We have updated our revision (Section 4.4) based on your comments and colored by red. We address each comment in detail, one by one as below.
>
> **(Q1) The soft-labeling scheme is counter-intuitive.**
>
> (A1) It is important to understand that the nature of our problem setup already assumes that there is no correct way to label a given out-of-class sample. Our soft-labeling scheme can be rather viewed as a more reasonable way to label such out-of-class samples compared to existing state-of-the-art semi-supervised learning (SSL) methods, e.g., MixMatch [1] simply assigns its sharpened predictions. Our extensive experimental results further verify the effectiveness of this scheme.
>
> We also remark that assigning soft-labels to novel data is known to be beneficial in other contexts, e.g., transfer learning [2]. Similarly, our soft-labeling could give informative signals for in-classes. To further support this, we have included two supporting results in the revision: namely, we show (i) minimizing only the cross-entropy loss with soft-labels (without in-class samples) from scratch results in non-trivial (much better than guessing) classification performance (Section 4.4), i.e., our soft-labeling scheme indeed extracts useful labels, and (ii) the actual soft-labels assigned are qualitatively capturing meaningful information of in-classes well (Figure 3 in Section 4.4).
>
> We hope that our above response clarify your concerns.
>
> **(Q2) “The originality of the approach is limited.”**
>
> (A2) The key novelty of our paper is not in a particular design of each component of OpenCoS, but in the following two surprising observations: (i) out-of-class samples, i.e., samples that cannot be categorized with a closed set of labels, can improve the in-class representations via soft-labels, and (ii) contrastive representation is effective for detecting such out-of-class samples.
> We would like to emphasize that our work is much useful under a **realistic**, yet under-explored SSL scenario in the literature, which we also believe an important contribution for the important problem (when other researchers pursue similar tasks in the future).
>
>
> **(Q3) The ablation study shows that most of the improvement comes from a more accurate detection of out-of-class samples?**
>
> (A3) We first emphasize that the accurate detection scheme of out-of-class samples itself is also one of our key contributions: e.g., although RealMix also integrates such a detection scheme upon MixMatch [1], its detection performance is much worse than ours.
>
> Furthermore, we point out our ablation study also clearly confirms the effectiveness of the proposed auxiliary loss and BN: overall, they show a consistent improvement in all the benchmarks tested, and sometimes it is significant, e.g., at ImageNet benchmarks in Table 3. In other words, we verify an important message that even out-of-class samples could be beneficial for SSL from the results.
>
> -----------------------------------------------------------------------------------------------------------------------------------------
>
> [1] Berthelot et al., “MixMatch: A Holistic Approach to Semi-Supervised Learning”, NeurIPS 2019.
>
> [2] Li et al., “Learning without Forgetting”, ECCV, 2016.

---

> > ### Author Response · Authors · 2020-11-18
> > **Response to R2 (2/2)**
> >
> > **(Q4) Open-set semi-supervised learning baselines.**
> >
> > (A4) We first clarify that our original submission did compare with two recent approaches in open-set semi-supervised learning, namely Uncertainty-Aware Self-Distillation (UASD) [3] and RealMix [4], those attempt to detect and filter out the out-of-class samples. In particular, we remark RealMix is a baseline that does integrate an out-of-class detection scheme into MixMatch, similar to what you suggested. We highlighted them in the revision.
> >
> > Moreover, as suggested by R3, we have added a new baseline for open-set semi-supervised learning, namely DS3L [5], in Table 1 of the revision. Our method consistently outperforms DS3L on the overall benchmarks; for example, it achieves 56.32%, compared to 68.37% of ours in the CIFAR-10 + TinyImageNet benchmark.
> >
> > In summary, we have three baselines, UASD [3], RealMix [4] and DS3L [5], which handles open-set semi-supervised learning scenarios. We hope this clarifies your concern.
> >
> >
> > **(Q5) Why does our detection score (with the projection header) perform better than standard out-of-distribution detection scores?**
> >
> > (A5) We first clarify that our problem setup is completely different to the standard out-of-distribution (OOD) detection: OOD detection targets *unseen* out-of-distribution samples in test time, while our setup aims to detect *seen* out-of-class samples during training from few information of in-class labels. Hence, it is quite unclear whether existing OOD methods work for our problem. Nevertheless, we apply them as reported in Table 10 (Appendix B.3), and found that the standard out-of-distribution techniques could not perform well in our SSL setup having a small number (4 or 25 per class) of in-class labeled data, i.e., the softmax probability is not well-trained enough to detect OOD well under such a small data regime. We have further clarified this in Section 4.4 of the revision.
> >
> > On the other hand, regarding the improvements from the projection header: we agree that it is hard to prove why projection header is beneficial on detecting out-of-class samples, but there are some empirical observations supporting it (including ours in this work): e.g., one of recent work on OOD detection [6] also reports that using the projection header is beneficial when contrastive representation is used for the standard OOD detection. We think analyzing the more role of projection header in contrastive learning would be an interesting future work.
> >
> >
> > **Minor comment:**
> >
> > We thank the reviewer for pointing out a typo. The typo is corrected.
> >
> > -----------------------------------------------------------------------------------------------------------------------------------------
> >
> > [3] Chen et al., “Semi-supervised learning under class distribution mismatch”, AAAI 2020.
> >
> > [4] Nair et al., “Realmix: Towards realistic semi-supervised deep learning algorithms”, arXiv 2019.
> >
> > [5] Guo et al., “Safe deep semi-supervised learning for unseen-class unlabeled data”, ICML 2020.
> >
> > [6] Tack et al., “CSI: Novelty detection via contrastive learning on distributionally shifted instances”, NeurIPS2020.

---

### Official Review · AnonReviewer4 · 2020-10-27
**Solid work but marginal novelty/technical contributions**

**Rating:** 5
**Confidence:** 5

**Review:**

The authors proposed to address the task of semi-supervised learning (SSL) by contrastive learning techniques, and the proposed techniques can be applied to handle open-set unlabeled data (i.e., the label spaces between label and unlabeled data are partially disjoint). I found the paper clearly written and easy to follow. The review comments are discussed below.

The proposed learning scheme is extended from SimCLR by Chen et al. More precisely, the authors perform unsupervised learning using D_l (w/o observing labels) and D_u using SimCLR techniques. In order to handle out-of-class samples in D_u, the authors present the idea of learning class-wise prototypical representation based on the above contrastive features. Detection of such samples is performed by a simple cosine similarity comparison between each instance in D_u and the prototypes of D_l. The main contribution lies in the last stage, i.e., SSL with auxiliary loss, which trains classifiers for recognizing in-class samples while assigning soft label scores for out-of-sample ones. The use of such soft labels allows the training of such unlabeled and out-of-class samples, which would be the major novelty of this work. (I do not feel that the use of batch norm would be viewed as technical contributions.)

Overall, I feel that most of the technical components come from existing works (e.g., SimCLR from Chen et al. 2020, auxiliary batch norm from Xie et al. 2020.). The use of soft-label assignments has also been proposed by existing works, which makes the overall technical contributions to be marginal. The lack of verification on the soft label assignment would be the concern as well. For example, is "cat" assigned with 0.1*leopard + 0.2*lion + 0.7*tiger? Most importantly, are out-of-class samples assigned with uniformly soft labels (i.e., 1/C)? Out-of-class samples might still exhibit similarity with selected in-class categories, and thus forcing their soft labels to be a uniform distribution might not seem to be practical (if that's the case). Existing vision and learning models have been considering class-similarity based representation, etc. techniques for handling zero-shot or open-dataset learning problems (e.g., Xian et al. PAMI'18 and Scheirer et al. PAMI'12). Based on the above remarks, I feel that the paper is yet above the ICLR standard for acceptance.

---

> ### Author Response · Authors · 2020-11-18
> **Response to R4**
>
> Dear R4,
>
> We sincerely appreciate your constructive feedback to help us improve the manuscript. We have updated our revision (Section 4.4) based on your comments and colored by red. We address each comment in detail, one by one as below.
>
> **(Q1) The lack of verification on the soft-labeling scheme.**
>
> (A1)  It is important to understand that the nature of our problem setup already assumes that there is no correct way to label a given out-of-class sample. Our soft-labeling scheme can be rather viewed as a more reasonable way to label such out-of-class samples compared to existing state-of-the-art SSL methods, e.g., MixMatch [1] simply assigns its sharpened predictions. Our extensive experimental results further verify the effectiveness of this scheme.
>
> We also remark that assigning soft-labels to novel data is known to be beneficial in other contexts, e.g., transfer learning [2]. Similarly, our soft-labeling could give informative signals for in-classes. To further support this, we have included two supporting results in the revision: namely, we show (i) minimizing only the cross-entropy loss (with soft-labels) from scratch results in non-trivial (much better than guessing) classification performance (Section 4.4), i.e., our soft-labeling scheme indeed extracts useful labels, and (ii) the actual soft-labels assigned are qualitatively capturing meaningful information of in-classes well (Figure 3 in Section 4.4).
>
> We hope that our above response clarify your concerns.
>
>
> **(Q2) The overall technical contributions are marginal.**
>
> (A2) We believe the key contribution of our work is not in a particular design of each technical component, but rather in the following two surprising observations: (i) out-of-class samples, i.e., samples that cannot be categorized with a closed set of labels, can improve the in-class representations via soft-labels, and (ii) contrastive representation is effective for detecting such out-of-class samples. Our general framework of OpenCoS empirically verifies these observations, compared to existing semi-supervised learning methods that struggle under the presence of out-of-class samples. We would like to emphasize that our work is much useful under a **realistic**, yet under-explored SSL scenario in the literature, which we also believe an important contribution for the important problem (when other researchers pursue similar tasks in the future).
>
> -----------------------------------------------------------------------------------------------------------------------------------------
>
> [1] Berthelot et al., “MixMatch: A Holistic Approach to Semi-Supervised Learning”, NeurIPS 2019.
>
> [2] Li et al., “Learning without Forgetting”, ECCV, 2016.

---

### Official Review · AnonReviewer1 · 2020-10-28
**Review of OpenCoS**

**Rating:** 4
**Confidence:** 5

**Review:**

This paper considers the problem of semi-supervised learning, where the unlabeled data may include out-of-class samples. To address this task, the paper proposes a method consisting of three steps: (1) detecting out-of-class samples in the unlabeled set, (2) assigning soft-labels to the detected out-of-class samples using class-conditional likelihoods from labeled data, and (3) using auxiliary batch normalization layers  to help mitigate the class distribution mismatch problem. Experiments are conducted on CIFAR-10, CIFAR-100, ImageNet datasets. Results show improvements over competing methods.

Quality: This paper is well written and well organized. I find this paper easy to follow.

Clarify: The preliminaries section clearly describes the setting of semi-supervised learning concerned in this paper and also clearly describes the contrastive representation learning. I like the way that the authors include a preliminaries section before the proposed method section. This allows me to understand which parts of the method are built upon existing approaches and helps me better identify the contributions and new things of this paper.

Originality: While I think the idea of this paper makes sense, I think there are several parts that are highly similar from a recent paper published in ECCV 2020 [a]. While the setting considered in [a] (for metric learning problems) is a bit different from that concerned in this submission (for image classification tasks), the high-level idea (learning representations that can be used to describe unlabeled images with labels different from those in the training set) is very similar. For example, the idea of Equation 5 and Equation 6 in this submission is almost exactly the same as that in Equation 2 of [a]. In addition, the idea of Equation 9 of this submission is very similar to that of Equation 3 of [a]. However, the authors did not acknowledge the similarity with [a] in the submission. This will make readers feel like the idea of Equation 5 and Equation 6 is original in this paper.

[a] Chen et al. Learning to learn in a semi-supervised fashion. In ECCV, 2020. https://arxiv.org/pdf/2008.11203.pdf

Significance: Given that some parts of this paper are highly similar to [a], the significance of this paper is downplayed.

Request for author response: I would like to see how the authors compare their paper with [a] in terms of problem setting, idea of class-wise similarity and representation learning. In particular, I would like to know the pros and cons of this paper compared to [a]. It would be great if the authors can talk about whether the semantics-oriented similarity representation in [a] could be used (and how to use it) to help improve the performance of the proposed method in the setting concerned by the paper.

Rating: Given that there are many parts similar to [a] and they are not acknowledged in the paper, I can only rate 4 for this submission at this point. I will reevaluate this paper after seeing the reviews from the other reviews as well as the author response.

---

> ### Author Response · Authors · 2020-11-18
> **Response to R1**
>
> Dear R1,
>
> We sincerely appreciate your feedback and for introducing a related work of “Learning to Learn in a Semi-Supervised Fashion” [1]. We have updated our revision (Section 3.2) based on your comments and colored by red. We address each comment in detail, one by one as below.
>
> **(Q1) Comparison with [1].**
>
> (A1) Many thanks for your suggestion to compare with [1]. At a high level, both [1] and our work consider unlabeled dataset containing out-of-class samples, and systematically define (soft) labels of them based on the class-wise similarity that also has been adopted in several prior works [1, 2, 3]. However, we do believe that [1] is incomparable to ours due to the following reasons:
> - Our goal is to improve the classification performance on the in-classes using the out-of-class samples, while [1] aims to learn a representation that can generalize to novel out-of-classes (at test time) using other out-of-classes as unlabeled samples during training. In other words, the evaluation goal is completely different, i.e., we measure the performance of in-class, while [1] does that of out-class.
> - Due to the different goals, the actual soft-labeling schemes used in ours and [1] are also different in terms of the representation used: OpenCoS is based on a contrastive representation learned from the (labeled and unlabeled) dataset, while [1] considers a meta-learning on the labeled samples aiming to transfer better at (unseen) out-of-class samples.
> - Finally, we remark our framework uses the class-wise similarity not only to define soft labels, but also to explicitly identify the out-of-class samples on-the-fly during training, which was not addressed in [1].
>
> Instead, we rather think DS3L [4] that suggested by R3 can be a proper additional baseline to compare, and we have added the comparison in Table 1 of the revision: our method still consistently and significantly outperforms it on all the benchmarks tested.
>
>
> **(Q2) Joint use of OpenCoS and the semantics-oriented similarity representation [1]?**
>
> (A2) Thanks for your suggestion. We emphasize that our work highlights the benefit of unsupervised representations in (semi-) supervised learning: such a label-free representation turns out to enhance model generalization due to its robustness on the novel, out-of-class samples. Hence, although we focus on SimCLR in this paper (due to its recent success in unsupervised learning), we expect our framework of OpenCoS can be applied to any other advanced pre-trained representation, e.g., the semantics-oriented similarity representation [1] as you suggested, which would be an interesting future work to explore.
>
> -----------------------------------------------------------------------------------------------------------------------------------------
>
> [1] Chen et al., “Learning to Learn in a Semi-Supervised Fashion”, ECCV 2020.
>
> [2] Vinyals et al., “Matching Networks for One Shot Learning”, NeurIPS 2016.
>
> [3] Snell et al., “Prototypical Networks for Few-shot Learning”, NeurIPS 2017.
>
> [4] Guo et al., “Safe deep semi-supervised learning for unseen-class unlabeled data”, ICML 2020.

---

### Official Review · AnonReviewer3 · 2020-10-29
**Review for the paper**

**Rating:** 7
**Confidence:** 4

**Review:**

##########################################################################
Summary:

The paper proposes a new approach for open set semi-supervised learning, where there are unlabeled data from classes not in the labeled data. The paper uses a contrastive representation learning paradigm to learn a feature encoder and a similarity measurement. Then the paper filters outlier samples by the similarity measurement and further utilizes outlier samples with soft labels. The separate BN layers address the distribution shift between in-class and out-class data.

##########################################################################

Reasons for score:


Overall, I vote for accepting. I like the idea of learning representation in an unsupervised way for both labeled and unlabeled data. My major concern is about how to select a reasonable threshold and why the threshold works well (see cons below). Hopefully the authors can address my concern in the rebuttal period.

##########################################################################

Pros:

The paper addresses a very interesting and practical problem in semi-supervised learning, where the unlabeled samples may include out-class samples. This problem promotes the practical use of semi-supervised learning in real world applications.

The paper employs contrastive representation learning to learn the encoder and similarity measurement, which preserves the similarity between features of a sample transformed by different transformations. The learned encoder can encode the semantic information into the feature for both labeled and unlabeled data in an unsupervised way.

The paper filters the out-class samples by the learned similarity measure based on a threshold, which can better filter out out-class samples.

The paper employs different batch normalization layers for the in-class and out-class samples, which avoids the influence of the distribution shift.

The paper is well-written and the claims are clearly clarified.

Experimental results on three semi-supervised learning benchmarks: CIFAR-10, CIFAR-100 and ImageNet, show that the proposed method can detect open-class samples and achieves higher accuracy the previous semi-supervised learning methods. Qualitative results show that the method is not very sensitive to hyper-parameters.

##########################################################################

Cons:

The performance gain of the proposed method is only over closed-set semi-supervised learning methods. The paper does not compare with the state-of-the-art open set semi-supervised learning method (Guo et al., 2020).

The threshold hyper-parameter is the key to filtering out-class samples. The paper decides the threshold by mean minus 2 standard deviation. Could the authors explain why choosing this value? Is there any insight in this threshold or why does this threshold fitful for any dataset?

---

> ### Author Response · Authors · 2020-11-18
> **Response to R3**
>
> Dear R3,
>
> We sincerely appreciate your valuable comments and efforts to help us improve the manuscript. We have updated our revision (Section 4.2) based on your comments and colored by red. We address each comment in detail, one by one as below.
>
> **(Q1) Open-set semi-supervised learning baselines.**
>
> (A1) We first clarify that our original submission did compare two recent approaches in open-set semi-supervised learning, namely Uncertainty-Aware Self-Distillation (UASD) [1] and RealMix [2]. We highlighted them again in the revision. Following your suggestion, we have further compared our method with DS3L [3] in Table 1 of the revision. We could not include the results of DS3L in Table 2 of the current revision mainly due to our resource constraint, but will definitely include the complete results in the final draft: we note a full run of DS3L for Table 2 takes  ∼200 GPU hours in our machine.
>
> As summarized in Table 1 below, DS3L is not a strong baseline, e.g., DS3L often underperforms the simple baseline, SimCLR-le, and our method still consistently outperforms it on all the benchmarks tested in Table 1; e.g., it achieves 56.32%, compared to 68.37% of ours in the CIFAR-10 + TinyImageNet benchmark.
>
> | **In-class**               | **CIFAR-Animals**   |       **CIFAR-10**      |  **CIFAR-10**  | **CIFAR-100**     | **CIFAR-100**|
> |:-------------------------    |:-------------------------------:   | :------------------------------:    | :----------------------------:    | :----------------------:    | :----------------------:    |
> | **Out-of-class**           | **CIFAR-Others**  |        **SVHN**          | **TinyImageNet**  | **SVHN**          | **TinyImageNet**  |
> |            |   | **# labels per class = 4**          |   |           |   |
> | SimCLR-le                      | 65.58 | 56.89 | 58.20 | 22.86 | 27.93 |
> | SimCLR-ft                      | 67.29 | 42.16 | 54.26 | 18.99 | 29.57 |
> | UASD-ft                        | 43.92 | 42.99 | 50.38 | 19.66 | 25.72 |
> | RealMix-ft                     | 64.42 | 38.22 | 48.28 | 18.48 | 22.14 |
> | DS3L-ft                        | 63.98 | 36.81 | 56.32 | 16.35 | 23.95 |
> | MixMatch-ft                    | 44.34 | 23.71 | 38.90 | 13.45 | 23.16 |
> | FixMatch-ft                    | 34.94 | 32.70 | 35.99 | 23.56 | 30.70 |
> | ReMixMatch-ft                  | 47.61 | 24.56 | 28.51 |  9.36 | 22.33 |
> | + OpenCoS (ours) | 77.66 | 61.33 | 68.37 | 28.43 | 36.51 |
> |            |   | **# labels per class = 25**          |   |           |   |
> | SimCLR-le                      | 80.03 | 70.31 | 71.84 | 37.74 | 43.68 |
> | SimCLR-ft                      | 81.44 | 64.41 | 73.05 | 39.61 | 49.69 |
> | UASD-ft                        | 82.17 | 66.70 | 73.97 | 39.51 | 44.58 |
> | RealMix-ft                     | 80.27 | 58.15 | 69.19 | 44.14 | 47.57 |
> | DS3L-ft                        | 81.31 | 50.00 | 69.13 | 29.00 | 40.16 |
> | MixMatch-ft                    | 83.88 | 17.98 | 69.27 | 38.60 | 50.23 |
> | FixMatch-ft                    | 69.86 | 68.02 | 70.49 | 41.73 | 45.94 |
> | ReMixMatch-ft                  | 81.62 | 37.98 | 67.38 | 32.75 | 49.63 |
> | + OpenCoS (ours) | 86.89 | 78.84 | 82.46 | 49.02 | 54.09 |
>
>
> **(Q2) Detection threshold hyperparameter.**
>
> (A2) We empirically observe our choice of the detection threshold $\mu - 2 \sigma$ generally performs well for all the dataset tested in our experiments. Assuming that the score follows Gaussian assumption, this threshold would give the 95% confidence, which is a common yet reasonable choice (Appendix B.4). Nevertheless, we also remark that more tuning could further improve the results, e.g., we observe that $\mu - 3 \sigma$ performs better in the CIFAR-Animal + CIFAR-Others benchmark (Appendix B.4).
>
> -----------------------------------------------------------------------------------------------------------------------------------------
>
> [1] Chen et al., “Semi-supervised learning under class distribution mismatch”, AAAI 2020.
>
> [2] Nair et al., “Realmix: Towards realistic semi-supervised deep learning algorithms”, arXiv 2019.
>
> [3] Guo et al., “Safe deep semi-supervised learning for unseen-class unlabeled data”, ICML 2020.

---

### Author Response · Authors · 2020-11-19
**Summary of Revisions**

Dear reviewers,


We sincerely appreciate your insightful comments and constructive suggestions to help us improve the manuscript. We are grateful for all positive comments: well-written (by R1, R2, R3 and R4), easy-to-follow (by R1, R4), clearly clarified (by R1, R3), and extensive experimental setup and qualitative results (by R3).
In response to the questions and concerns you raised, we have carefully revised and enhanced the manuscript with the following additional experiments and discussions:


- Highlighting open-set SSL baselines we compared (Section 4.2, Appendix A.2),
- Performance comparisons with a new open-set SSL baseline, DS3L [1] (Section 4.2),
- More clarification on our detection task and method (Section 4.4),
- Additional ablation study of the proposed soft-label assignment (Section 4.4, Figure 3), and its detailed experimental setups (Appendix A.2)

The revisions made are marked with “red” in the revised manuscript.


Best regards,

Authors.


[1] Guo et al., “Safe deep semi-supervised learning for unseen-class unlabeled data”, ICML 2020.

---

### Decision · Program_Chairs · 2021-01-07
**Final Decision**

**Decision:**

Reject

**Comment:**

This paper proposes OpenCos for semi-supervised learning that can leverage unsupervised information in open-set scenarios where samples can be out-of-class.  They first pre-train by learning an unsupervised representation using SimCLR on both the labeled and unlabeled data.  Then, they detect out-of-class samples in the unlabeled set based on similarity measures on the representation learned in the previous step.  The unlabeled data can now be split into in-class and out-of-class.  OpenCos optimizes (8) which combines a semi-supervised loss for in-class unlabeled data and an auxiliary cross-entropy loss with soft-labels for the out-of-class samples.  Finally, they perform an auxiliary batch normalization.

The paper is easy to read and clearly structured.  It also places the work well with respect to related work.

The proposed approach makes sense; however, as pointed out by the reviewers, the novelty is marginal.  The technical innovation seems to be an extension of SimCLR and the auxiliary batch norm of Xie et al.